# Achieving PAC Guarantees in Mechanism Design through Multi-Armed Bandits

**Takayuki Osogami**[*]  *osogami@jp.ibm.com*
*IBM Research – Tokyo*
**Hirota Kinoshita**[*]  *hirotak@ttic.edu*
*Toyota Technological Institute at Chicago*[†]
**Segev Wasserkrug**  *segevw@il.ibm.com*
*IBM Research – Israel*

**Reviewed on OpenReview:** *https://openreview.net/forum?id=tbe8143jO8*

## Abstract

We analytically derive a class of optimal solutions to a linear program (LP) for automated mechanism design that satisfies efficiency, incentive compatibility, strong budget balance (SBB), and individual rationality (IR), where SBB and IR are enforced in expectation. Our solutions can be expressed using a set of essential variables whose cardinality is exponentially smaller than the total number of variables in the original formulation. However, evaluating a key term in the solutions requires exponentially many optimization steps as the number of players $N$ increases. We address this computational bottleneck by formulating it as a best mean estimation problem in the multi-armed bandit (MAB) framework, where the goal is pure exploration to estimate expectations rather than online learning with exploration-exploitation tradeoffs. We develop a probably approximately correct (PAC) estimator with asymptotically optimal sample complexity. This MAB-based statistical estimation approach reduces the optimization complexity from exponential to $O(N \log N)$. Numerical experiments confirm that our method efficiently computes mechanisms with the target properties, scaling to problems with up to $N = 128$ players—substantially improving over prior work.

## 1 Introduction

Mediators in multi-agent systems can improve efficiency by making centralized decisions that maximize social welfare. In trading networks, for example, they ensure goods are produced by the lowest-cost firms and allocated to those with the greatest need (Hatfield et al., 2013; Osogami et al., 2023). While such mediators could prioritize profit by charging participants—like today's tech giants operating digital marketplaces or advertising platforms—they often capture most of the surplus, leaving participants with little gain.

We instead envision an open platform designed to maximize benefits for participants in multi-agent systems. This aligns with the goals of auction mechanisms for *public* resources, where resources should go to those who value them most, and the mediator should run without budget deficits or surpluses (Bailey, 1997; Cavallo, 2006; Dufton et al., 2021; Gujar & Narahari, 2011; Guo, 2012; Guo & Conitzer, 2009; Manisha et al., 2018; Tacchetti et al., 2022). However, such mechanisms rely on the structure of single-sided auctions, where all participants are buyers. In contrast, similar guarantees may be unattainable in more general settings—such as double-sided auctions (Hobbs et al., 2000; Zou, 2009; Widmer & Leukel, 2016; Stößer et al., 2010; Kumar et al., 2018; Chichin et al., 2017), matching markets (Zhang & Zhu, 2020), or trading networks (Osogami et al., 2023; Wasserkrug & Osogami, 2023)—even when they are possible in the single-sided case.

---

[*]Equal contributions
[†]This work was performed at IBM Research

We design mechanisms for general environments encompassing all the above multi-agent systems, focusing on efficiency and strong budget balance (SBB). Specifically, the mediator selects a social decision that maximizes the total value to the players (i.e., decision efficiency; DE), while ensuring its expected revenue equals a target $\rho \in \mathbb{R}$ (SBB when $\rho = 0$). Following standard Bayesian mechanism design (Shoham & Leyton-Brown, 2009), each player's valuation depends on their private type, while the joint distribution of types is common knowledge. Since DE is hard to achieve without knowing types, we require dominant strategy incentive compatibility (DSIC): truth-telling must be optimal regardless of others' strategies. To promote participation, we also impose individual rationality (IR): each player's expected utility must exceed a type-dependent threshold $\theta(t_n) \in \mathbb{R}$ (with standard IR when $\theta \equiv 0$).

While these properties are standard in mechanism design (Shoham & Leyton-Brown, 2009), we introduce parameters, $\rho$ and $\theta$, to generalize the standard definitions of SBB and IR, motivated by three considerations. First, the standard definitions can make it impossible to satisfy all four desired properties simultaneously (Green & Laffont, 1977; Myerson & Satterthwaite, 1983; Osogami et al., 2023); our generalization enables precise characterization of when they are achievable. Second, it supports a principled, sample-based mechanism learning approach with theoretical guarantees, allowing the resulting mechanisms to approximately satisfy the four properties with high probability. Third, the parameters help model practical constraints—e.g., a mediator may require positive revenue to sustain the platform or need to guarantee players positive utility to attract participation.

We require SBB and IR to hold *in expectation* (specifically, *ex ante* and *interim*, respectively) over the distribution of types, while DE and DSIC must hold for every realization of types (*ex post*). These assumptions align with those in Osogami et al. (2023); Wasserkrug & Osogami (2023) for trading networks, though the *in expectation* requirements are clearly weaker than the *ex post* guarantees typically assumed in auction settings (Bailey, 1997; Cavallo, 2006; Dufton et al., 2021; Gujar & Narahari, 2011; Guo, 2012; Guo & Conitzer, 2009; Manisha et al., 2018; Tacchetti et al., 2022). However, this relaxation allows us to derive *analytical* solutions for mechanisms that satisfy all four desired properties in general environments and to characterize when such mechanisms exist. In contrast, prior analytical results are limited to auctions with a single good type (Bailey, 1997; Cavallo, 2006; Guo, 2011; Guo & Conitzer, 2007; 2009; Moulin, 2009) or unit-demand settings (Gujar & Narahari, 2011; Guo, 2012). For more complex auctions (Dufton et al., 2021; Manisha et al., 2018; Tacchetti et al., 2022) and trading networks (Osogami et al., 2023; Wasserkrug & Osogami, 2023), mechanisms are computed via numerical optimization, which often scales poorly with the number of players.

In particular, while the theoretical framework in Osogami et al. (2023) could, in principle, be applied to any finite number of players, their numerical method is constrained by computational complexity and has been applied only to trading networks with *two* players. In contrast, our analytical solutions can be evaluated numerically for around 10 players, depending on the number of types. The computational bottleneck in our analytical solutions is evaluating the minimum expected value over possible types for each player. Doing so exactly requires computing an efficient social decision for all $K^N$ type profiles, where $K$ is the number of possible types per player and $N$ is the number of players.

To address this bottleneck, we formulate the evaluation of the minimum expected value as a best mean estimation (BME) problem in the multi-armed bandit (MAB) framework (Lattimore & Szepesvári, 2020). Crucially, our use of MAB differs from typical applications: rather than online learning with exploration-exploitation tradeoffs (as in regret minimization (Auer et al., 1995; 2002)) or finding the best arm (as in best arm identification (Audibert et al., 2010; Maron & Moore, 1993; Mnih et al., 2008; Bubeck et al., 2009)), we employ MAB purely as a statistical estimation tool for computing expectations efficiently. Specifically, our goal is to estimate the mean reward of the best arm—a pure exploration task without sequential decision-making or exploitation. We develop a probably approximately correct (PAC) algorithm that estimates this value within $\varepsilon$ error with probability at least $1-\delta$, and show that its sample complexity, $O((K/\varepsilon^2) \log(1/\delta))$, matches our derived lower bound. This MAB-based statistical estimation approach reduces the number of computations of efficient social decisions from $K^N$ to $O(KN \log N)$, allowing us to handle cases up to 128 players—a substantial improvement over two players in Osogami et al. (2023).

**Our Results**

We address the problem of designing mechanisms for general multi-agent environments that satisfy four desired properties: DE, DSIC, $\theta$-IR, and $\rho$-SBB. Our main contributions consist of analytical characterization, computational efficiency, and PAC guarantees.

**Analytical Characterization.** We provide necessary and sufficient conditions for the existence of mechanisms satisfying these four desired properties:

**Theorem 1** (Informal version of Lemmas 1 and 2)**.** *The LP for computing a mechanism satisfying DE, DSIC, $\theta$-IR, and $\rho$-SBB admits feasible solutions if and only if condition* (10) *holds (necessity proven under independent type distributions). Under this condition, there exists an optimal solution that has an analytic expression involving minimum expectation terms (*$\min_{t_n \in \mathcal{T}_n} \mathbb{E}[w^\star(t) \mid t_n]$*).*

This characterization reduces the number of essential variables in the LP from $N K^{N-1}$ (exponential) to $N$ (linear), where $N$ is the number of players and $K$ is the number of types per player.

**Computational Efficiency via Multi-Armed Bandits.** Evaluating the analytical solution requires computing $\min_{t_n \in \mathcal{T}_n} \mathbb{E}[w^\star(t) \mid t_n]$ for each player $n$, which naively requires $K^N$ evaluations. We formulate this as a BME problem in MAB and provide PAC guarantees:

**Theorem 2** (Informal version of Theorems 3 and 4)**.** *There exists an $(\varepsilon, \delta)$-PAC algorithm that computes a mechanism satisfying DE, DSIC, $\theta$-IR, and $\rho$-SBB with probability at least $1-\delta$, using $O((NK/\varepsilon^2)\log(N/\delta))$ evaluations of efficient decision $w^\star(t)$.*

This approach reduces computational complexity from exponential ($K^N$) to nearly linear in the number of players, enabling scalability to 128 players with 1024 types.

**Organization.** Section 2 reviews related work. Sections 3–4 formulate the problem. Section 5 provides analytical characterization. Sections 6–7 develop the MAB-based approach with PAC guarantees. Section 8 presents empirical validation.

## 2 Related Work

The most closely related prior work is Osogami et al. (2023), which formulates and numerically solves an LP to design mechanisms for trading networks that satisfy DE, DSIC, IR, and weak budget balance (WBB), where the mediator's expected revenue is nonnegative. While the LP in Osogami et al. (2023) uses an arbitrary objective and includes SBB only as an example, our work focuses specifically on SBB and derives analytical solutions tailored to this objective. We also extend their formulation with tunable parameters.

The remainder of this section reviews related work on mechanism design—particularly efforts to achieve SBB—and on MAB, focusing on PAC algorithms. Notably, our approach is unique in estimating a key quantity from our analytically derived optimal solution, leading to the proposal of an asymptotically optimal PAC algorithm for the underexplored objective of best mean estimation.

In single-sided auctions where only buyers act strategically, Vickrey–Clarke–Groves (VCG) mechanisms with Clarke's pivot rule (VCG auctions) satisfy *ex post* DE (often called allocative efficiency in auctions), DSIC, IR, and WBB (Nisan, 2007). However, the Green–Laffont Impossibility Theorem shows that no mechanism can achieve DE, DSIC, and SBB in all environments (Green & Laffont, 1977; 1979). This has led to efforts to redistribute the mediator's revenue to players while maintaining DSIC, DE, IR, and WBB. Analytical mechanisms have been derived for auctions with single or homogeneous goods (Bailey, 1997; Cavallo, 2006; Guo, 2011; Guo & Conitzer, 2007; 2009; Moulin, 2009), or unit-demand bidders (Gujar & Narahari, 2011; Guo, 2012). For auctions with multi-unit demands for heterogeneous goods, prior work has proposed numerical methods to approximate optimal redistribution, using piecewise linear functions (Dufton et al., 2021) or neural networks (Manisha et al., 2018; Tacchetti et al., 2022).

We consider general environments with heterogeneous goods and multi-unit demands, where players may have both negative and positive valuations for social decisions (e.g., players may buy or sell goods depending

on the social decision). Under these conditions, the Myerson-Satterthwaite Impossibility Theorem (Myerson & Satterthwaite, 1983) shows that no mechanism can guarantee *ex post* DE, DSIC[1], IR, and WBB in general environments, unlike the VCG auctions in the single-sided environment. Thus, we derive mechanisms that achieve DE, DSIC, IR, and SBB as best as possible by introducing parameters $\theta$ and $\rho$ in IR and SBB. A limitation of our results is that IR and SBB hold only in expectation, but this is justifiable for risk-neutral mediators and players (Osogami et al., 2023). Additionally, our model can guarantee *strictly* positive expected utility, which can ensure nonnegative utility with high probability when players repeatedly participate in the mechanism.

Much of the literature focuses on maximizing the mediator's revenue in auctions (Myerson, 1981), with recent work on automated mechanism design (AMD) using machine learning (Duetting et al., 2019; Rahme et al., 2021; Ivanov et al., 2022; Curry et al., 2020) and analyzing sample complexity (Balcan et al., 2016; Morgenstern & Roughgarden, 2015; Syrgkanis, 2017). Similar to these studies, we formulate an optimization problem whose solution yields a mechanism with the desired properties. However, rather than solving it numerically, we derive optimal solutions analytically. While prior work analyzes the sample complexity for maximizing expected revenue, we focus on the sample complexity of evaluating specific expressions in analytically derived mechanisms.

We evaluate our analytical expression using best mean estimation (BME) in MAB, where standard objectives include regret minimization (Auer et al., 1995; 2002) and best arm identification (BAI) (Audibert et al., 2010; Maron & Moore, 1993; Mnih et al., 2008; Bubeck et al., 2009). The most relevant prior work on MAB is PAC learning for BAI and its sample complexity analysis. We reduce BME to BAI and establish a lower bound on BME's sample complexity using a technique from BAI (Even-Dar et al., 2002). While this method does not give tight bounds for BAI (Mannor & Tsitsiklis, 2004), it provides tight bounds for BME. Note that the problem of estimating the best mean frequently appears in reinforcement learning (van Hasselt, 2010) and machine learning (Kajino et al., 2023), where the focus is on estimating the best mean from a given set of samples (van Hasselt, 2013), while our focus is on efficiently collecting samples for the estimation.

## 3 Settings

The goal of mechanism design is to specify the rules of a game in a way that an outcome desired by the mechanism designer is achieved when rational players, aiming to maximize their individual utility, participate (Jackson, 2014; Shoham & Leyton-Brown, 2009). Let $\mathcal{N} := [N] := \{1, 2, \ldots, N\}$ be the set of players and $\mathcal{O}$ be the set of possible outcomes. For each player $n \in \mathcal{N}$, let $\mathcal{A}_n$ be the set of available actions and $\mathcal{T}_n$ be the set of possible types. Let $\mathcal{A} := \mathcal{A}_1 \times \ldots \times \mathcal{A}_N$ and $\mathcal{T} := \mathcal{T}_1 \times \ldots \times \mathcal{T}_N$ be the corresponding product spaces. A mechanism $\mu : \mathcal{A} \to \mathcal{O}$ determines an outcome depending on the actions taken by the players. Let $u_n : \mathcal{O} \times \mathcal{T}_n \to \mathbb{R}$ be the utility function of each player $n \in \mathcal{N}$.

We consider Bayesian games where the players' types follow a probability distribution that is known to all players and the mediator. Before selecting actions, the players know their own types but not the types of the other players. A strategy of each player $n \in \mathcal{N}$ is thus a function from $\mathcal{T}_n$ to $\mathcal{A}_n$.

We assume that an outcome is determined by a social decision and payment; hence, a mechanism $\mu$ consists of a decision rule and a payment rule. Let $\mathcal{D}$ be the set of possible social decisions. Given the actions of the players, the decision rule $\phi : \mathcal{A} \to \mathcal{D}$ determines a social decision, and the payment rule $\tau : \mathcal{A} \to \mathbb{R}^{\mathcal{N}}$ determines the amount of (possibly negative) payment to the mediator from each player. Let $v : \mathcal{D} \times (\mathcal{T}_1 \cup \ldots \cup \mathcal{T}_N) \to \mathbb{R}$ specify the value of a given social decision to the player of a given type. Then the utility of player $i$ when players take actions $a \in \mathcal{A}$ is

$$u_n(\mu(a); t_n) = u_n((\phi(a), \tau(a)); t_n) = v(\phi(a); t_n) - \tau_n(a).$$

We assume that $\mathcal{N}$, $\mathcal{D}$, and $\mathcal{T}_n, \forall n \in \mathcal{N}$ are finite sets.

Without loss of generality by the revelation principle (Shoham & Leyton-Brown, 2009), we consider only direct mechanisms, where the action available to each player is to declare their type to from the set of possible types (i.e., $\mathcal{A}_n = \mathcal{T}_n, \forall n \in \mathcal{N}$). We thus use $\mathcal{T}_n$ for $\mathcal{A}_n$.

---

[1](Myerson & Satterthwaite, 1983) shows stronger results with Bayesian-Nash Incentive Compatibility.

Then we seek to achieve the following four properties with our mechanisms. The first property is Dominant Strategy Incentive Compatibility (DSIC), which ensures that the optimal strategy of each player is to truthfully reveal its type regardless of the strategies of the other players. Formally,

$$[\text{DSIC}] \quad v(\phi(t_n, t'_{-n}); t_n) - \tau_n(t_n, t'_{-n}) \geq v(\phi(t'); t_n) - \tau_n(t'), \forall t' \in \mathcal{T}, \forall t_n \in \mathcal{T}_n, \forall n \in \mathcal{N},$$

where the left-hand side represents the utility of the player having type $t_n$ when it declares the same $t_n$, and the other players declare arbitrary types $t'_{-n}$.

The second property is Decision Efficiency (DE), which requires that the mediator chooses the social decision that maximizes the total value to the players. With DSIC, we can assume that the players declare true types, and hence we can write DE as a condition on the decision rule:

$$[\text{DE}] \qquad \phi(t) \in \underset{d \in \mathcal{D}}{\operatorname{argmax}} \sum_{n \in \mathcal{N}} v(d; t_n) \qquad \forall t \in \mathcal{T}.$$

As the third property, we generalize individual rationality and require that the expected utility of each player is at least as large as a target value that can depend on its type. We refer to this property as $\theta$-IR. Again, assuming that players declare true types due to DSIC, we can write $\theta$-IR as follows:

$$[\theta\text{-IR}] \quad \mathbb{E}[v(\phi(t); t_n) - \tau_n(t) \mid t_n] \geq \theta(t_n) \qquad \forall t_n \in \mathcal{T}_n, \forall n \in \mathcal{N},$$

where $\theta : \mathcal{T}_1 \cup \ldots \cup \mathcal{T}_N \to \mathbb{R}$ determines the target expected utility for each type. Throughout (except in Section 6, where we discuss general MAB models), $\mathbb{E}$ denotes the expectation with respect to the probability distribution $\mathbb{P}$ of types, which is the only probability that appears in our mechanisms.

The last property is a generalization of Budget Balance (BB), which we refer to as $\rho$-WBB and $\rho$-SBB. Specifically, $\rho$-WBB requires that the expected revenue of the mediator is no less than a given constant $\rho \in \mathbb{R}$, and $\rho$-SBB requires that it is equal to $\rho$. Again, assuming that the players declare true types due to DSIC, these properties can be written as follows:

$$[\rho\text{-WBB}] \qquad \sum_{n \in \mathcal{N}} \mathbb{E}\left[\tau_n(t)\right] \geq \rho. \qquad\qquad [\rho\text{-SBB}] \qquad \sum_{n \in \mathcal{N}} \mathbb{E}\left[\tau_n(t)\right] = \rho.$$

While $\rho$-SBB is stronger than $\rho$-WBB, we will see that $\rho$-SBB is satisfiable if and only if $\rho$-WBB is satisfiable.

## 4 Optimization Problem for Automated Mechanism Design

Following Osogami et al. (2023), we seek to find optimal mechanisms in the class of VCG mechanisms[2], specified by a pair $(\phi^\star, h)$. Specifically, after letting players take the actions of declaring their types $t \in \mathcal{T}$, the mechanism first finds a social decision $\phi^\star(t)$ using a decision rule $\phi = \phi^\star$ that satisfies DE. It then determines the amount of payment from each player $n \in \mathcal{N}$ to the mediator by

$$\tau_n(t) = h_n(t_{-n}) - \sum_{m \in \mathcal{N}_{-n}} v(\phi^\star(t); t_m), \tag{1}$$

where we define $\mathcal{N}_{-n} := \mathcal{N} \setminus \{n\}$, and $h_n : \mathcal{T}_{-n} \to \mathbb{R}$ is an arbitrary function of the types of the players other than $n$ and referred to as a pivot rule. The decision rule $\phi^\star$ guarantees DE by construction, and the payment rule (1) then guarantees DSIC (Nisan, 2007).

Our problem is now reduced to find the pivot rule, $h = \{h_n\}_{n \in \mathcal{N}}$, that minimizes the expected revenue of the mediator, while satisfying $\theta$-IR and $\rho$-WBB. This may lead to satisfying $\rho$-SBB if the revenue is maximally reduced. To represent this reduced problem, let

$$w^\star(t) := \sum_{n \in \mathcal{N}} v(\phi^\star(t); t_n) \tag{2}$$

---

[2]Since our setting involves finite type spaces, VCG mechanisms may not exhaust all mechanisms satisfying the four properties (DE, DSIC, $\theta$-IR, $\rho$-SBB) for a given $(\theta, \rho)$. Nevertheless, we restrict attention to VCG mechanisms, as they are both of theoretical interest in the literature and of practically importance with various real-world implementations. Focusing on this principled class provides a tractable path to analytical solutions while maintaining strong theoretical guarantees.

be the total value of the efficient social decision when the players have types $t$. Then we can rewrite $\theta$-IR (for the player having type $t_n$) and $\rho$-WBB as follows:

$$\mathbb{E}[v(\phi^\star(t); t_n) - \tau_n(t) \mid t_n] \geq \theta(t_n) \iff \mathbb{E}\left[\sum_{m \in \mathcal{N}} v(\phi^\star(t); t_m) \,\middle|\, t_n\right] - \mathbb{E}[h_n(t_{-n}) \mid t_n] \geq \theta(t_n) \tag{3}$$

$$\iff \mathbb{E}[w^\star(t) \mid t_n] - \mathbb{E}[h_n(t_{-n}) \mid t_n] \geq \theta(t_n) \tag{4}$$

and

$$\sum_{n \in \mathcal{N}} \mathbb{E}[\tau_n(t)] \geq \rho \iff \sum_{n \in \mathcal{N}} \mathbb{E}[h_n(t_{-n})] - \sum_{n \in \mathcal{N}} \sum_{m \in \mathcal{N}_{-n}} \mathbb{E}[v(\phi^\star(t); t_m)] \geq \rho \tag{5}$$

$$\iff \sum_{n \in \mathcal{N}} \mathbb{E}[h_n(t_{-n})] - (N-1)\mathbb{E}[w^\star(t)] \geq \rho, \tag{6}$$

where the last equivalence follows from the definition of $w^\star(t)$ in (2).

Therefore, we arrive at the following linear program (LP):

$$\min_h \quad \sum_{n \in \mathcal{N}} \mathbb{E}[h_n(t_{-n})] \tag{7}$$

$$\text{s.t.} \quad \mathbb{E}[w^\star(t) \mid t_n] - \mathbb{E}[h_n(t_{-n}) \mid t_n] \geq \theta(t_n) \qquad \forall t_n \in \mathcal{T}_n, \forall n \in \mathcal{N} \tag{8}$$

$$\sum_{n \in \mathcal{N}} \mathbb{E}[h_n(t_{-n})] - (N-1)\mathbb{E}[w^\star(t)] \geq \rho. \tag{9}$$

The approach in Osogami et al. (2023) solves this LP numerically (considering only the case with $\rho = 0$ and $\theta \equiv 0$). Since there is one variable $h_n(t_{-n})$ for each $t_{-n} \in \mathcal{T}_{-n}$ and $n \in \mathcal{N}$, the LP involves $N K^{N-1}$ variables and $NK + 1$ constraints when each player has $K$ possible types.

If the LP is feasible, let $h^\star$ denote its optimal solution; the resulting VCG mechanism $(\phi^\star, h^\star)$ then satisfies DSIC, DE, $\theta$-IR, and $\rho$-WBB. Formally, the following proposition holds:

**Proposition 1.** *Let the decision rule $\phi^\star$ be the one that satisfies DE and the payment rule $\tau = (\tau_n)_n$ be in the form of (1) where $h = (h_n)_n = (h_n^\star)_n = h^\star$ is given by the solution to the LP (7)-(9). Then the VCG mechanism $(\phi^\star, \tau^\star)$ satisfies DSIC, DE, $\theta$-IR, and $\rho$-WBB.*

*Proof.* With the equivalences (4)-(6), the constraints (8)-(9) in the LP guarantee that $\theta$-IR and $\rho$-WBB are satisfied by feasible solutions. Since we consider the class of VCG mechanisms, DE is trivially satisfied by the definition of $\phi^\star$, and DSIC is satisfied when the payment rule is in the form of (1). Hence, all of DSIC, DE, $\theta$-IR, and $\rho$-WBB are satisfied by $(\phi^\star, \theta^\star)$. $\qquad\square$

If the LP is infeasible, no such VCG mechanism exists. In Section 5, we fully characterize the feasibility condition and derive analytical solutions.

For clarity, in Algorithm 1, we summarize the protocol under the VCG mechanism. In Step 3, the optimal strategy of each player is to truthfully declare its type $\hat{t}_n = t_n$. In Step 5, the LP may not be feasible, in which case the protocol may fail, or we may use another payment rule to proceed.

## 5 Analytical Solutions to the Optimization Problem

We first establish a sufficient condition and a necessary condition for the LP to have feasible solutions.

**Lemma 1.** *The LP given by (7)-(9) is feasible if*

$$\sum_{n \in \mathcal{N}} \min_{t_n \in \mathcal{T}_n} \{\mathbb{E}[w^\star(t) \mid t_n] - \theta(t_n)\} \geq (N-1)\mathbb{E}[w^\star(t)] + \rho. \tag{10}$$

*When types are independent ($t_m$ and $t_n$ are independent for any $m \neq n$ under $\mathbb{P}$), the LP is feasible if and only if (10) holds. When types are dependent, the LP may be feasible even if (10) is violated.*

---

**Algorithm 1** Protocol under the VCG mechanism

1: Sample the type profile $t$ from the common prior $\mathbb{P}$
2: Each player $n \in \mathcal{N}$ gets to know its own type $t_n$
3: Each player $n \in \mathcal{N}$ declares their type $\hat{t}_n$
4: Determine the social decision: $\phi^\star(\hat{t}) \leftarrow \underset{d \in \mathcal{D}}{\operatorname{argmax}} \sum_{n \in \mathcal{N}} v(d; \hat{t}_n)$
5: $h^\star \leftarrow$ Find the optimal solution to the LP (7)-(9)
6: Determine the payment from each player $n \in \mathcal{N}$ to the mediator:
$$\tau_n(\hat{t}) \leftarrow h_n^\star(\hat{t}_{-n}) - \sum_{m \in \mathcal{N}_{-n}} v(\phi^\star(\hat{t}); \hat{t}_m)$$
7: Each player $n \in \mathcal{N}$ gets utility $v(\phi^\star(\hat{t}); t_n) - \tau_n(\hat{t})$

---

*Proof.* We start by establishing the sufficiency of (10) for the feasibility of the LP (7)-(9), regardless of whether the types are independent or not. Specifically, we will show that

$$h_n(t_{-n}) = \eta_n := \min_{t_n' \in \mathcal{T}_n} \left\{ \mathbb{E}[w^\star(t') \mid t_n'] - \theta(t_n') \right\} \qquad \forall t_{-n} \in \mathcal{T}_{-n} \tag{11}$$

is a feasible solution when (10) holds. Hence, $h_n(t_{-n})$ in (11) will turn out to be independent of $t_{-n}$.

The $\theta$-IR (8) is satisfied with (11), because for any $t_n \in \mathcal{T}_n$ and $n \in \mathcal{N}$ we have

$$\mathbb{E}[h_n(t_{-n}) \mid t_n] = \eta_n = \min_{t_n' \in \mathcal{T}_n} \left\{ \mathbb{E}[w^\star(t') \mid t_n'] - \theta(t_n') \right\} \leq \mathbb{E}[w^\star(t) \mid t_n] - \theta(t_n). \tag{12}$$

The $\rho$-WBB (9) is satisfied with (11), because

$$\sum_{n \in \mathcal{N}} \mathbb{E}[h_n(t_{-n})] = \sum_{n \in \mathcal{N}} \eta_n = \sum_{n \in \mathcal{N}} \min_{t_n \in \mathcal{T}_n} \left\{ \mathbb{E}[w^\star(t) \mid t_n] - \theta(t_n) \right\} \geq (N-1)\,\mathbb{E}[w^\star(t)] + \rho, \tag{13}$$

where the inequality follows from (10). This establishes the sufficiency of (10).

Next, we prove the necessity of (10) for the feasibility of the LP (7)-(9) when the types are independent. When the types are independent, (8) is reduced to

$$\mathbb{E}[w^\star(t) \mid t_n] - \mathbb{E}[h_n(t_{-n})] \geq \theta(t_n) \qquad \forall t_n \in \mathcal{T}_n, \forall n \in \mathcal{N} \tag{14}$$

$$\iff \min_{t_n \in \mathcal{T}_n} \left\{ \mathbb{E}[w^\star(t) \mid t_n] - \theta(t_n) \right\} \geq \mathbb{E}[h_n(t_{-n})] \qquad \forall n \in \mathcal{N}. \tag{15}$$

This together with (9) establishes the necessity of

$$(N-1)\,\mathbb{E}[w^\star(t)] + \rho \leq \sum_{n \in \mathcal{N}} \mathbb{E}[h_n(t_{-n})] \tag{16}$$

$$\leq \sum_{n \in \mathcal{N}} \min_{t_n \in \mathcal{T}_n} \left\{ \mathbb{E}[w^\star(t) \mid t_n] - \theta(t_n) \right\}. \tag{17}$$

Finally, we construct an example that satisfies (8)-(9) but violates (10). Let $\mathcal{N} = \{1, 2\}$; $\mathcal{T}_n = \mathcal{T} := \{1, 2\}, \forall n \in \mathcal{N}$; $\rho = 0$; $\theta(m) = 0, \forall m \in \mathcal{T}$. We assume that the types are completely dependent (namely, $t_1 = t_2$ surely) and let $p$ be the probability that $t_1 = t_2 = 1$ (hence, $t_1 = t_2 = 2$ with probability $1 - p$).

For this example, we rewrite (8)-(9) and (10) by using $x_m := w^\star((m, m))$ and $y_{nm} := h_n(m)$ for $m \in \mathcal{T}$ and $n \in \mathcal{N}$. Notice that, for any $m \in \mathcal{T}$ and $n \in \mathcal{N}$, we have $\mathbb{E}[w^\star(t) \mid t_n = m] = x_m$ and $\mathbb{E}[h_n(t_{-n}) \mid t_n = m] = y_{nm}$, since types are completely dependent. Hence, (8) is reduced to

$$x_m - y_{nm} \geq 0 \qquad \forall m \in \mathcal{T}, \forall n \in \mathcal{N} \tag{18}$$

and (9) is reduced to

$$p\,(y_{11} + y_{21} - x_1) + (1 - p)\,(y_{12} + y_{22} - x_2) \geq 0. \tag{19}$$

On the other hand, (10) is reduced to

$$2 \min\{x_1, x_2\} \geq p \, x_1 + (1 - p) \, x_2. \tag{20}$$

Consider the case where $x_m > 0, \forall m \in \mathcal{T}$. In this case, (18)-(19) suggest that (8)-(9) are satisfied as long as $y_{nm}$ satisfies

$$\frac{x_m}{2} \leq y_{nm} \leq x_m \qquad \forall m \in \mathcal{T}, \forall n \in \mathcal{N}, \tag{21}$$

whether (20) is satisfied or not. Indeed, (21) can be met even if (20) is violated, for example when $p = \frac{1}{2}, x_1 = 1, x_2 = 4, y_{nm} = \frac{2}{3}x_m, \forall m \in \mathcal{T}, \forall n \in \mathcal{N}$; this serves as a desired example, concluding the proof. $\square$

The proof of Lemma 1 offers intuition behind the condition (10), which arises as the requirement for a constant pivot rule to satisfy $\theta$-IR and $\rho$-WBB. A key insight of our results is that this sufficient condition is also necessary when player types are independent.

Building on this, we analytically derive a class of *optimal* solutions to the LP under condition (10), regardless of whether types are independent or not:

**Lemma 2.** *A pivot rule is called constant if and only if there exists a constant $\eta_n$ such that $h_n(t_{-n}) = \eta_n, \forall t_{-n} \in \mathcal{T}_{-n}$ for each $n \in \mathcal{N}$. Let $\mathcal{H}$ be the set of constant pivot rules with:*

$$\eta_n = \min_{t_n \in \mathcal{T}_n} \{\mathbb{E}[w^\star(t) \mid t_n] - \theta(t_n)\} - \delta_n \qquad \forall n \in \mathcal{N}, \tag{22}$$

*where $(\delta_n)_{n \in \mathcal{N}}$ lies on the following simplex:*

$$\delta_n \geq 0 \qquad \forall n \in \mathcal{N} \tag{23}$$

$$\sum_{n \in \mathcal{N}} \delta_n = \sum_{n \in \mathcal{N}} \min_{t_n \in \mathcal{T}_n} \{\mathbb{E}[w^\star(t) \mid t_n] - \theta(t_n)\} - (N - 1) \, \mathbb{E}[w^\star(t)] - \rho. \tag{24}$$

*Then, any $h \in \mathcal{H}$ is an optimal solution to the LP (7)-(9). Furthermore, $\mathcal{H}$ is nonempty if and only if (10) holds.*

*Proof.* We first rewrite the LP (7)-(9) in the following equivalent form:

$$\min_{h} \quad \sum_{n \in \mathcal{N}} \sum_{t_n \in \mathcal{T}_n} \mathbb{P}[t_n] \, \mathbb{E}[h_n(t_{-n}) \mid t_n] \tag{25}$$

$$\text{s.t.} \quad \mathbb{E}[w^\star(t) \mid t_n] - \mathbb{E}[h_n(t_{-n}) \mid t_n] \geq \theta(t_n) \qquad \forall t_n \in \mathcal{T}_n, \forall n \in \mathcal{N} \tag{26}$$

$$\sum_{n \in \mathcal{N}} \sum_{t_n \in \mathcal{T}_n} \mathbb{P}[t_n] \, \mathbb{E}[h_n(t_{-n}) \mid t_n] - (N - 1) \, \mathbb{E}[w^\star(t)] \geq \rho. \tag{27}$$

Hence, to prove that any $h \in \mathcal{H}$ is an optimal solution, it suffices to show for any $h \in \mathcal{H}$ that (27) is satisfied with equality and (26) is satisfied. When $h \in \mathcal{H}$, we have, for any $t_n$, that

$$\mathbb{E}[h_n(t_{-n}) \mid t_n] = \eta_n \leq \min_{t'_n \in \mathcal{T}_n} \{\mathbb{E}[w^\star(t) \mid t'_n] - \theta(t'_n)\} \leq \mathbb{E}[w^\star(t) \mid t_n] - \theta(t_n), \tag{28}$$

where the first inequality follows from (23). We also have

$$\sum_{n \in \mathcal{N}} \sum_{t_n \in \mathcal{T}_n} \mathbb{P}[t_n] \, \mathbb{E}[h_n(t_{-n}) \mid t_n] = \sum_{n \in \mathcal{N}} \sum_{t_n \in \mathcal{T}_n} \mathbb{P}[t_n] \, \eta_n = \sum_{n \in \mathcal{N}} \eta_n = (N - 1) \, \mathbb{E}[w^\star(t)] + \rho, \tag{29}$$

where the last equality follows from (22) and (24).

Next, $\mathcal{H}$ is nonempty when (10) holds, because the following $\eta_n$ satisfies the conditions (22)-(24):

$$\eta_n = \min_{t_n \in \mathcal{T}_n} \{\mathbb{E}[w^\star(t) \mid t_n] - \theta(t_n)\} - \delta \tag{30}$$

where

$$\delta := \frac{1}{N} \left( \sum_{n \in \mathcal{N}} \min_{t_n \in \mathcal{T}_n} \{\mathbb{E}[w^\star(t) \mid t_n] - \theta(t_n)\} - (N-1)\,\mathbb{E}[w^\star(t)] - \rho \right). \tag{31}$$

Notice that $\delta \geq 0$ follows from (10).

Finally, (10) is necessary for $\mathcal{H}$ to be nonempty, since if (10) does not hold, we have

$$\sum_{n \in \mathcal{N}} \min_{t_n \in \mathcal{T}_n} \{\mathbb{E}[w^\star(t) \mid t_n] - \theta(t_n)\} - (N-1)\,\mathbb{E}[w^\star(t)] - \rho < 0, \tag{32}$$

which makes $\mathcal{H}$ empty. □

In deriving the optimal solutions, we have substantially reduced the essential number of variables (from $N K^{N-1}$ to $N$ when each player has $K$ types). Our approach can therefore not only find but also represent or store solutions exponentially more efficiently than Osogami et al. (2023). Moreover, it turns out that the solutions in $\mathcal{H}$ not only satisfy $\rho$-WBB but also $\rho$-SBB (in addition to DE, DSIC, and $\theta$-IR) regardless of whether the types are independent or not. Formally, the following corollary holds:

**Corollary 1.** *Any VCG mechanism given with a pivot rule in $\mathcal{H}$ satisfies $\rho$-SBB.*

*Proof of Corollary 1.* By (22), we have

$$\sum_{n \in \mathcal{N}} \eta_n - (N-1)\,\mathbb{E}[w^\star(t)] - \rho = \sum_{n \in \mathcal{N}} \min_{t_n \in \mathcal{T}_n} \{\mathbb{E}[w^\star(t) \mid t_n] - \theta(t_n)\} - \sum_{n \in \mathcal{N}} \delta_n - (N-1)\,\mathbb{E}[w^\star(t)] - \rho \tag{33}$$

$$= 0, \tag{34}$$

where the last equality follows from (24). Hence, (9) is satisfied with equality. □

When types are independent, the condition (10) is necessary for the existence of a feasible solution; hence, we do not lose optimality by considering only the solutions in $\mathcal{H}$. When types are dependent, the condition (10) may still be satisfied, and *the solutions in $\mathcal{H}$ remain optimal in this case.* However, when types are dependent, the LP may be feasible even if (10) is violated, and in this case optimal solutions are not in the space of constant pivot rules. In the proof of Lemma 1, we construct such a case with an extreme example of completely dependent types. However, (10) is often satisfied even in such extreme cases of completely dependent types. For example, as long as

$$x_1 \leq x_2 \leq \frac{2-p}{1-p}\,x_1, \tag{35}$$

condition (10) is satisfied in the example in the proof of Lemma 1, since then $(x_1, x_2)$ satisfies (20), which corresponds to (10) in this example.

When the LP is infeasible, we may construct a mechanism that satisfies one of $\rho$-SBB and $\theta$-IR (in addition to DE and DSIC) regardless of whether the types are independent or not. Specifically, any VCG mechanism with a pivot rule that satisfies (22) and (24) ensures $\rho$-SBB, and any VCG mechanism with a pivot rule that satisfies (22) and (23) ensures $\theta$-IR for any $t_n \in \mathcal{T}_n$ and $n \in \mathcal{N}$. Formally, the following corollary holds:

**Corollary 2.** *Any VCG mechanism with a pivot rule that satisfies* (22) *and* (24) *ensures $\rho$-SBB. Any VCG mechanism with a pivot rule that satisfies* (22) *and* (23) *ensures $\theta$-IR for any $t_n \in \mathcal{T}_n$ and $n \in \mathcal{N}$.*

*Proof of Corollary 2.* The first part of the corollary can be proved analogously to Corollary 1. Regarding the second part of the corollary, by (22), for any $t_n \in \mathcal{T}_n$ and $n \in \mathcal{N}$, we have

$$\mathbb{E}[w^\star(t) \mid t_n] - \eta_n - \theta(t_n) = \mathbb{E}[w^\star(t) \mid t_n] - \theta(t_n) - \left( \min_{t'_n \in \mathcal{T}_n} \{\mathbb{E}[w^\star(t) \mid t'_n] - \theta(t'_n)\} \right) + \delta_n, \tag{36}$$

which is nonnegative by (23), and hence (8) holds. □

For example, for any $t_n \in \mathcal{T}_n$ and $n \in \mathcal{N}$, the following pivot rule always satisfies $\theta$-IR:

$$\eta_n = \min_{t_n \in \mathcal{T}_n} \left\{ \mathbb{E}[w^\star(t) \mid t_n] - \theta(t_n) \right\} - \max\{\delta, 0\} \qquad \forall n \in \mathcal{N}, \tag{37}$$

where $\delta$ is as defined in (31). Also, $\rho$-SBB is ensured by (30) with $\delta$ in (31).

Alternatively, one may choose $\theta$ and $\rho$ in a way that they ensure feasibility of the LP (i.e., (10) is satisfied). For example, the LP becomes feasible if we set $\theta \equiv 0$ and

$$\rho = \left[ \sum_{n \in \mathcal{N}} \min_{t_n \in \mathcal{T}_n} \mathbb{E}[w^\star(t) \mid t_n] - (N-1)\, \mathbb{E}[w^\star(t)] \right]^-, \tag{38}$$

where $[x]^- := \min\{x, 0\}$ for $x \in \mathbb{R}$. When $\rho < 0$, the mediator might get negative expected revenue, but the expected loss of the mediator is at most $|\rho|$.

One may also set $\rho = 0$ and

$$\theta(t_n) = \left[ \mathbb{E}[w^\star(t) \mid t_n] - \frac{N-1}{N} \mathbb{E}[w^\star(t)] \right]^- \qquad \forall t_n \in \mathcal{T}_n, \forall n \in \mathcal{N} \tag{39}$$

to guarantee the feasibility of the LP, since

$$\sum_{n \in \mathcal{N}} \min_{t_n \in \mathcal{T}_n} \left\{ \mathbb{E}[w \mid t_n] - \left[ \mathbb{E}[w^\star(t) \mid t_n] - \frac{N-1}{N} \mathbb{E}[w^\star(t)] \right]^- \right\} - (N-1)\, \mathbb{E}[w^\star(t)]$$

$$= \sum_{n \in \mathcal{N}} \left( \min_{t_n \in \mathcal{T}_n} \left\{ \mathbb{E}[w \mid t_n] - \frac{N-1}{N} \mathbb{E}[w^\star(t)] - \left[ \mathbb{E}[w^\star(t) \mid t_n] - \frac{N-1}{N} \mathbb{E}[w^\star(t)] \right]^- \right\} \right) \tag{40}$$

$$\geq 0. \tag{41}$$

In this case, player $n$ may incur negative utility when it has type $t_n$ with $\theta(t_n) < 0$, although the loss is guaranteed to be bounded by $|\theta(t_n)|$.

While our analytical solutions significantly reduce computational cost compared to the numerical approach in Osogami et al. (2023), they still require evaluating

$$\kappa_n(\theta) := \min_{t_n \in \mathcal{T}_n} \left\{ \mathbb{E}[w^\star(t) \mid t_n] - \theta(t_n) \right\} \tag{42}$$

for each $n \in \mathcal{N}$. Since $\mathbb{E}$ is the expectation over the distribution $\mathbb{P}$ on $\mathcal{T}$, this requires evaluating $w^\star(t)$ for all $t \in \mathcal{T}$. From (2), $w^\star(t)$ is the total value of the efficient decision $\phi^\star(t)$, which itself is the solution to an optimization problem defining DE. Without any structure in $\mathcal{D}$ or $v$, this requires evaluating the total value for every decision in $\mathcal{D}$.

## 6 Evaluating the Analytical Solutions with a PAC Guarantee

To reduce the computational cost of (42), we adopt a learning-based approach. The key observation is that estimating (42) can be cast as a variant of a multi-armed bandit (MAB) problem, where the goal is to estimate the mean reward of the best arm. Specifically, pulling an arm $t_n \in \mathcal{T}_n$ yields a reward of $\theta(t_n) - w^\star(t)$, with $t$ sampled from the conditional distribution $\mathbb{P}[\cdot \mid t_n]$. To be consistent with prior MAB literature (Even-Dar et al., 2002; 2006; Hassidim et al., 2020; Mannor & Tsitsiklis, 2004), we frame the problem as one of reward maximization.

Since we assume that $\mathcal{N}$, $\mathcal{D}$, and $\mathcal{T}_n, \forall n \in \mathcal{N}$ are finite, there exist constants, $\bar{\theta}$ and $\bar{v}$, such that $|\theta(t')| \leq \bar{\theta}$ and $|v(d; t')| \leq \bar{v}, \forall d \in \mathcal{D}, \forall t' \in \cup_{n \in \mathcal{N}} \mathcal{T}_n$. Then we can also bound $|\theta(t_n) - w^\star(t)| \leq \bar{\theta} + N\,\bar{v}, \forall t \in \mathcal{T}, \forall n \in \mathcal{N}$. Namely, the reward is bounded. We assume that the bounds are known, allowing us to scale the reward to the interval $[0, 1]$ surely.

We also assume that we have access to an arbitrary size of the sample that is independent and identically distributed (i.i.d.) according to $\mathbb{P}[\cdot \mid t_n]$ for any $t_n \in \mathcal{T}_n, n \in \mathcal{N}$. When players have independent types, such sample can be easily constructed as long as we have access to i.i.d. sample $\{t^{(i)}\}_{i=1,2,\dots}$ from $\mathcal{T}$, because $\{(t_n, t_{-n}^{(i)})\}_i$ is the sample from $\mathbb{P}[\cdot \mid t_n]$ for any $t_n \in \mathcal{T}_n, n \in \mathcal{N}$.

Consider the general $K$-armed bandit where each arm's reward is bounded in $[0, 1]$. For each $k \in [K]$, let $\mu_k$ be the true mean of arm $k$. Let $\mu_\star := \max_k \mu_k$ be the best mean-reward, which we seek to estimate. We say that the sample complexity of an algorithm for a MAB is $T$ if the algorithm pulls arms at most $T$ times.

A standard PAC algorithm for MAB returns an $\varepsilon$-optimal arm with probability at least $1 - \delta$ for given $\varepsilon, \delta$ (Even-Dar et al., 2006; Hassidim et al., 2020). On the other hand, we need to evaluate (42) within a given estimation error with high probability. Formally, we will use the following definitions:

**Definition 1.** *For $\varepsilon, \delta > 0$, we say that an algorithm is $(\varepsilon, \delta)$-PAC Best Arm Identifier (BAI) if the output $\hat{I}$ returned by the algorithm satisfies $\Pr\left(\mu_{\hat{I}} \geq \mu_\star - \varepsilon\right) \geq 1 - \delta$ and that an algorithm is $(\varepsilon, \delta)$-PAC Best Mean Estimator (BME) if the output $\hat{\mu}$ returned by the algorithm satisfies $\Pr\left(|\hat{\mu} - \mu_\star| \leq \varepsilon\right) \geq 1 - \delta$.*

BAI and BME are related but different[3].

We now show how PAC estimates of the analytical solution translate into approximate mechanism guarantees. The key result (Theorem 3) establishes that mechanisms computed using PAC BME satisfy all four desired properties (DE, DSIC, $\theta$-IR, $\rho$-WBB) with high probability. Readers primarily interested in the final algorithm and its guarantees may skip directly to Proposition 2, which summarizes the sample complexity result.

We can estimate the term $\kappa_n(\theta)$ in (42) by the use of an $(\varepsilon, \delta)$-PAC BME. The optimal solutions in Lemma 2 involves another term,

$$\lambda(\rho) := \mathbb{E}[w^\star(t)] + \rho/(N-1), \tag{43}$$

which can also be estimated using a standard PAC estimator of expectation. These allow us to find a mechanism that satisfies designed properties with high probability. Formally, recalling that $N$ is the number of players, we have the following lemma:

**Lemma 3.** *Let $\tilde{\kappa}_n(\theta)$ for $n \in \mathcal{N}$ and $\tilde{\lambda}(\rho)$ be independent estimates of $\kappa_n(\theta)$ and $\lambda(\rho)$ respectively given by an $(\varepsilon', \delta')$-PAC Best Mean Estimator and a standard $(\varepsilon'', \delta')$-PAC estimator of expectation. Also, let $\tilde{d} := d(\tilde{\kappa}(\theta), \tilde{\lambda}(\rho), \varepsilon''', \varepsilon'''')$ be a point on the following simplex (here, we change the notation from $\delta$ in Lemma 2 to $\tilde{d}$ to avoid confusion):*

$$\tilde{d}_n \geq \varepsilon''', \forall n \in \mathcal{N} \tag{44}$$

$$\sum_{n \in \mathcal{N}} \tilde{d}_n = \sum_{n \in \mathcal{N}} \tilde{\kappa}_n(\theta) - (N-1)\left(\tilde{\lambda}(\rho) + \varepsilon''''\right). \tag{45}$$

*Then the VCG mechanism with the constant pivot rule $h_n(t_{-n}) = \eta_n = \tilde{\kappa}_n(\theta) - \tilde{d}_n$ satisfies $(\theta - (\varepsilon' - \varepsilon'''))$-IR and $(\rho - (N-1)(\varepsilon'' - \varepsilon''''))$-WBB with probability $(1 - \delta')^{N+1}$.*

*Proof.* Note that the PAC estimators give independent estimates, $\tilde{\kappa}_n(\theta)$ for $n \in \mathcal{N}$ and $\tilde{\lambda}$, such that

$$\Pr\left(|\tilde{\kappa}_n(\theta) - \kappa_n(\theta)| \leq \varepsilon'\right) \geq 1 - \delta' \qquad \forall n \in \mathcal{N}, \tag{46}$$

$$\Pr\left(|\tilde{\lambda}(\rho) - \lambda(\rho)| \leq \varepsilon''\right) \geq 1 - \delta', \tag{47}$$

guaranteeing that $\Pr\left(E\right) \geq (1 - \delta')^{N+1}$, where $E := \{|\tilde{\lambda}(\rho) - \lambda(\rho)| \leq \varepsilon''\} \cap \bigcap_{n \in \mathcal{N}}\{|\tilde{\kappa}_n(\theta) - \kappa_n(\theta)| \leq \varepsilon'\}$.

---

[3]For example, consider a case where the best arm has large variance and $\mu_\star = 1/2$, and all the other arms always yield zero reward $\mu_n = 0, \forall n \neq \star$. Then BAI is relatively easy here due to the large gap $\mu_\star - \mu_n = 1/2, \forall n \neq \star$, while BME would demand more samples to accurately estimate $\mu_\star$ due to the large variance of the best arm. In contrast, suppose there are many arms with Bernoulli rewards: half have expected value 1, and the rest have expected value $1 - (3/2)\varepsilon$. By pulling sufficiently many arms (once for each arm), Hoeffding's inequality allows us to estimate that the best mean is at least $1 - \varepsilon$ with high probability—sufficient for BME. However, BAI would require identifying a specific arm with high expected value, necessitating many samples to distinguish it reliably (to be able to say that this particular arm has expected value at least $1 - \varepsilon$).

In the event $E$, the VCG mechanism with the constant pivot rule $h_n(t_{-n}) = \eta_n = \tilde{\kappa}_n(\theta) - \tilde{d}_n$ satisfies $(\theta - (\varepsilon' - \varepsilon'''))$-IR, because the expected utility of player $n$ given its type $t_n$ (recall (4)) is

$$\mathbb{E}[w^\star(t) \mid t_n] - (\tilde{\kappa}_n(\theta) - \tilde{d}_n) \geq \mathbb{E}[w^\star(t) \mid t_n] - \kappa_n(\theta) - \varepsilon' + \varepsilon''' \geq \theta(t_n) - (\varepsilon' - \varepsilon'''), \tag{48}$$

where the first inequality follows from the PAC bound on $\tilde{\kappa}_n(\theta)$ and the definition of $\tilde{d}_n$, and the last inequality follows from the original guarantee when $\kappa_n(\theta)$ is exactly computed. Meanwhile, $(\rho - (N-1)\,(\varepsilon'' - \varepsilon''''))$-WBB is also ensured in the event $E$, because the expected revenue of the mediator (recall (6)) is

$$\sum_{n \in \mathcal{N}} (\tilde{\kappa}_n(\theta) - \tilde{d}_n) - (N-1)\,\mathbb{E}[w^\star(t)] = (N-1)\,(\tilde{\lambda}(\rho) + \varepsilon''') - (N-1)\,\mathbb{E}[w^\star(t)] \tag{49}$$

$$\geq \rho + (N-1)\,(\varepsilon''' - \varepsilon'') \tag{50}$$

where the equality follows from the definition of $\tilde{d}$, and the inequality follows from the PAC bound on $\tilde{\lambda}(\rho)$. □

Notice that the sufficient condition of Lemma 1 states that the simplex in Lemma 2 is nonempty. Analogously, when the simplex in Lemma 3 is empty, we cannot provide the solution that guarantees the properties stated in Lemma 3. Since DSIC and DE remain satisfied regardless of whether $\kappa_n$ for $n \in \mathcal{N}$ and $\lambda$ are estimated or exactly computed, Lemma 3 immediately establishes the following theorem by appropriately choosing the parameters:

**Theorem 3.** *In Lemma 3, let $\varepsilon''' = \varepsilon'$, $\varepsilon'''' = \varepsilon''$, and $\delta' = 1 - (1-\delta)^{1/(N+1)}$. Then the VCG mechanism with the constant pivot rule $h_n(t_{-n}) = \tilde{\kappa}_n(\theta) - \tilde{d}_n$ satisfies DSIC, DE, $\theta$-IR, and $\rho$-WBB with probability $1 - \delta$.*

*Proof.* With the choice of the parameters in the theorem, we have $\theta - (\varepsilon' - \varepsilon''') = \theta$, $(\rho - (N-1)\,(\varepsilon'' - \varepsilon'''')) = \rho$, and $(1 - \delta')^{1/(N+1)} = 1 - \delta$. Hence, Lemma 3 guarantees that the constant pivot rule $h_n(t_{-n}) = \tilde{\kappa}_n(\theta) - \tilde{d}_n$ satisfies DSIC, DE, $\theta$-IR, and $\rho$-WBB with probability $1 - \delta$. □

Recall that computing our mechanism from Lemma 2 requires evaluating $w^\star(t)$ for all $t \in \mathcal{T}$, which grows exponentially with the number of players $N$. In Section 7, we show that there exists an $(\varepsilon, \delta)$-PAC BME with sample complexity $O((K/\varepsilon^2)\log(1/\delta))$, which can be used to reduce the number of required evaluations of $w^\star(t)$ to $O(N \log N)$, as is formally proved in the following proposition:

**Proposition 2.** *The sample complexity to learn the constant pivot rule in Theorem 3 is $O((N\,K/\varepsilon^2)\log(N/\delta))$, where $N = |\mathcal{N}|$ is the number of players, and $K = \max_{n \in \mathcal{N}} \mathcal{T}_n$ is the maximum number of possible types of each player.*

*Proof.* The constant pivot rule in Theorem 3 can be learned with $N$ independent runs of an $(\varepsilon, \delta')$-PAC BME and a single run of an $(\varepsilon, \delta')$-PAC estimator for an expectation, whose overall sample complexity is $O(N\,(K/\varepsilon^2)\log(1/\delta'))$. The proposition can then be established by substituting $\delta' = 1 - (1-\delta)^{1/(N+1)}$. □

# 7 Estimating the Best Mean Reward in Multi-Armed Bandits

This section establishes matching lower and upper bounds on the sample complexity for $(\varepsilon, \delta)$-PAC BME:

**Theorem 4.** *There exists an $(\varepsilon, \delta)$-PAC BME with sample complexity $O((K/\varepsilon^2)\log(1/\delta))$, and any $(\varepsilon, \delta)$-PAC BME must have the sample complexity at least $\Omega((K/\varepsilon^2)\log(1/\delta))$.*

These bounds match those for $(\varepsilon, \delta)$-PAC BAI (Mannor & Tsitsiklis, 2004), indicating that BME and BAI share similar sample complexity, despite their differences discussed in Section 6.

Our upper bound is established by reducing BME to BAI. Suppose that we have access to an arbitrary $(\varepsilon, \delta)$-PAC BAI with sample complexity $M$. We can construct a $((3/2)\varepsilon, 2\delta)$-PAC BME by first running the $(\varepsilon, \delta)$-PAC BAI and then taking $m^\star$ samples from the arm $\hat{I}$ that is identified as the best to estimate its mean (see Algorithm 2).

When $m^\star$ is appropriately selected, the following lemma holds:

---

**Algorithm 2** Best Mean Estimator

---

1: $\hat{I} \leftarrow \texttt{PAC-BAI}(\varepsilon, \delta)$
2: Pull arm $\hat{I}$ for $m^\star$ times
3: Let $\hat{\mu}_{\hat{I}}$ be the average of the $m^\star$ samples
4: **return** $\hat{\mu}_{\hat{I}}$

---

**Lemma 4.** *When an $(\varepsilon, \delta)$-PAC BAI with sample complexity $M$ is used, Algorithm 2 has sample complexity $M + m^\star$, where $m^\star := \lceil (2/\varepsilon^2) \log(1.22/\delta) \rceil$, and returns $\hat{\mu}_{\hat{I}}$ with*

$$\Pr\left(|\hat{\mu}_{\hat{I}} - \mu_\star| > (3/2)\varepsilon\right) \le 2\,\delta. \tag{51}$$

We prove the lemma in Appendix A. Since there exists an $(\varepsilon, \delta)$-PAC BAI with sample complexity $O((K/\varepsilon^2) \log(1/\delta))$ (Mannor & Tsitsiklis, 2004), this establishes the upper bound in Theorem 4.

Our lower bound is established by showing that an arbitrary $(\varepsilon, \delta)$-PAC BME can be used to solve the problem of identifying whether a given coin is negatively or positively biased (precisely, $\varepsilon$-Biased Coin Problem of Definition 2 in Appendix A), for which any algorithm is known to require *expected* sample complexity at least $\Omega((1/\varepsilon^2) \log(1/\delta))$ to solve correctly with probability at least $1 - \delta$ (Chernoff, 1972; Even-Dar et al., 2002) (see Lemma 6 in Appendix A). The following lemma, together with Lemma 6, establishes the lower bound on the sample complexity in Theorem 4:

**Lemma 5.** *If there exists an $(\varepsilon/2, \delta/2)$-PAC BME with sample complexity $M$ for $K$-armed bandit, then there also exists an algorithm, having expected sample complexity $M/K$, that can solve the $\varepsilon$-Biased Coin Problem correctly with probability at least $1 - \delta$.*

We prove the lemma in Appendix A. The proof technique of reduction from the $\varepsilon$-Biased Coin Problem was also used in Even-Dar et al. (2002) to prove a lower bound on the sample complexity of BAI. What is interesting, however, is that the lower bound in Even-Dar et al. (2002) is not tight when $\delta < 1/K$, and a tight lower bound is later established by a different technique in Mannor & Tsitsiklis (2004). In contrast, our reduction gives a tight lower bound on the sample complexity of BME.

The difference in the derived lower bound stems from the following behavior of BME and BAI when all the arms have mean reward of $\alpha^-$ and hence are indistinguishable. The algorithm constructed in Even-Dar et al. (2002) determines that $B$ has mean $\alpha^+$ when either arm $i^+$ or arm $i^-$ is identified as the best arm. When the arms are indistinguishable, a PAC BAI would correctly identify each of the $K$ arms, including $i^+$ or $i^-$, as the best arm uniformly at random, which induces an error with probability $1/K$. On the other hand, the mean reward estimated by a PAC BME would be approximately correct with high probability, even when the arms are indistinguishable.

## 8 Numerical Experiments

In this section, we present numerical experiments to validate our approach and examine its limitations. Specifically, we address three questions: i) BME in Section 6 has asymptotically optimal sample complexity, but how well can we estimate the best mean when there are only a moderate number of arms? ii) How much does BME reduce evaluations of $w^\star(t)$? iii) How well do $\theta$-IR and $\rho$-SBB hold when $\kappa_n(\theta)$ in (42) is estimated via BME?

Our results show that BME reduces computational cost by several orders of magnitude for a moderate to large number of players (up to 128), though its benefit declines as the number of types grows (up to 1024). All experiments run in a cloud environment on a single CPU core with up to 66 GB of memory, without GPU acceleration (see Table 1 in Appendix B). The source code in supplementary material will be open-sourced upon acceptance.

## 8.1 Comparison of Best Mean Estimation Algorithms

For BAI, algorithms with asymptotically optimal sample complexity often perform poorly when the number of arms is moderate (Hassidim et al., 2020). Consequently, Approximate Best Arm (Hassidim et al., 2020), despite its optimal complexity, switches to Naive Elimination—an asymptotically suboptimal method—when the number of arms is below $10^5$ or after eliminating sufficiently many suboptimal arms. The BME algorithm in Section 6 also achieves asymptotically optimal complexity but relies on BAI, so its performance for moderate arm counts is not well captured by Theorem 4. Like BAI, BME benefits from algorithms that perform well in this regime.

---

**Algorithm 3** Successive Elimination for Best Mean Estimation (($\varepsilon, \delta$)-PAC SE-BME)

---
**Require:** $\varepsilon, \delta$
 1: Let $\mathcal{R} \leftarrow \{1, \ldots, K\}$ be the set of remaining arms
 2: Let $t \leftarrow 0; \alpha \leftarrow 1$
 3: **while** $\alpha > \varepsilon$ **do**
 4:      Pull each arm $k \in \mathcal{R}$ once and update the sample average $\hat{\mu}_k$
 5:      $t \leftarrow t + 1$
 6:      $\alpha \leftarrow \sqrt{\frac{1}{2t} \log\left(\frac{\pi^2 K t^2}{3\delta}\right)}$
 7:      **for all** $k \in \mathcal{R}$ **do**
 8:          Remove $k$ from $\mathcal{R}$ if $\max_{\ell \in \mathcal{R}} \hat{\mu}_\ell - \hat{\mu}_k \geq 2\alpha$
 9:      **end for**
10: **end while**
11: **return** $\max_{k \in \mathcal{R}} \hat{\mu}_k$

---

We compare four BME algorithms across varying numbers of arms $K$, accuracy parameters $\varepsilon$, and confidence levels $\delta$. All algorithms solve the same BME problem and are compared under identical parameter settings. The algorithms are (see more detailed explanation in Appendix B.2):

1. **SE-BME**: Successive Elimination tailored directly for BME (Algorithm 3).

2. **SE**: Successive Elimination for BAI (Algorithm 4 in Appendix B) combined with Algorithm 2 to adapt it for BME.

3. **UGapEc**: The UGapEc algorithm (Gabillon et al., 2012) for BAI combined with Algorithm 2.

4. **Simple**: BME that collects $m = \lceil (2\varepsilon^2)^{-1} \log(2K/\delta) \rceil$ samples from each arm independently, and returns the maximum.

Figure 1 shows total sample sizes for the four algorithms with $\varepsilon = 0.05$ and $\delta = 0.05$. Arms have Bernoulli rewards with equally spaced means ($\mu_k = (k - 0.5)/K$ for $k = 1, \ldots, K$). SE-BME consistently achieves the lowest sample complexity across all values of $K$, demonstrating the benefit of direct BME design over BAI-based adaptations. The Simple baseline performs competitively, particularly for small $K$, validating its $O(K/\varepsilon^2 \log(K/\delta))$ complexity as reasonable in practice. Both SE and UGapEc require more samples than SE-BME, with their performance gap widening as $K$ increases. Additional results for varying $\varepsilon$ and $\delta$ are provided in Figure 7 (Appendix B), showing consistent trends across parameter settings.

## 8.2 Effectiveness of Best Mean Estimation in Mechanism Design

In this section, we quantitatively assess the effectiveness of BME in reducing the number of evaluations of $w^\star(t)$ when computing (42). We use ($\varepsilon, \delta$)-PAC SE-BME (Algorithm 3) as BME.

Our testbed is a mechanism design setting inspired by double-sided electricity auctions (Zou, 2009; Hobbs et al., 2000). We consider $N$ players, each with $K$ possible types, and vary $N$ and $K$. Each player's $K$ types are drawn uniformly without replacement from integers in $[-K, K]$. The common prior $\mathbb{P}$ assumes each player's type is uniformly distributed over its $K$ types and independent of others. A player acts as a

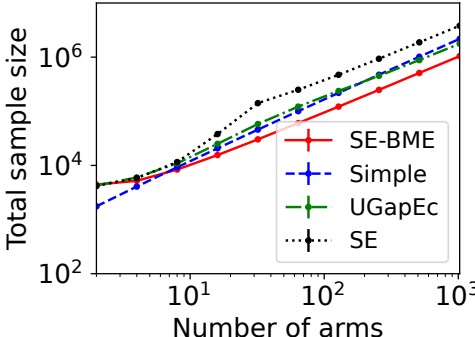

Figure 1: Total sample sizes required by four BME algorithms for Bernoulli rewards with equally spaced means ($\mu_k = (k - 0.5)/K$ for $k = 1, \ldots, K$) with $\varepsilon = 0.05$ and $\delta = 0.05$. Each point averages 10 runs; standard deviations are too small to be visible.

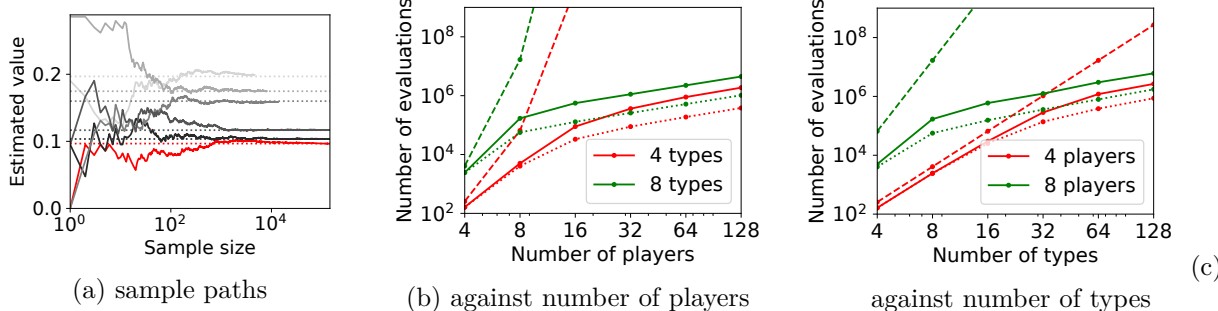

(a) sample paths       (b) against number of players       against number of types

Figure 2: (a) Representative sample paths of $\mathbb{E}[w^\star(t) \mid t_n]$ estimates for $t_n \in \mathcal{T}_n$ versus sample size under $(0.01, 0.1)$-PAC SE-BME for $N = K = 6$. (b)–(c) Unique evaluations of $w^\star(t)$ required to compute $\min_{t_n \in \mathcal{T}_n} \mathbb{E}[w^\star(t) \mid t_n]$ for all $n \in \mathcal{N}$: exact computation (dashed) vs. $(0.05, 0.05)$-PAC SE-BME (solid) and Simple BME (dotted).

buyer if its type $t_n > 0$ (valuation $v(d; t_n) = t_n$ for buying one unit) and as a seller if $t_n < 0$ (cost $|t_n|$ for selling one unit). Players with $t_n = 0$ do not participate. A social decision is a bipartite matching between buyers and sellers. In this setting, the social decision with DE can be computed by greedily matching buyers with high valuations to sellers with low costs, as long as buyer valuations exceed seller costs. We use bandit algorithms to solve the minimization problem in (42) (instead of the maximization problem as in Section 8.1), normalizing $w^\star(t)$ to the range $[0, 1]$.

Figure 2(a) shows sample paths of $\mathbb{E}[w^\star(t) \mid t_n]$ estimates for each $t_n \in \mathcal{T}_n$ using $(0.01, 0.1)$-PAC SE-BME to compute $\min_{t_n \in \mathcal{T}_n} \mathbb{E}[w^\star(t) \mid t_n]$ (i.e., (42) with $\theta(t_n) = 0$). Each panel displays $|\mathcal{T}_n|$ curves, with the red curve indicating the minimum. The number of players is fixed at $N = 6$ here and varied in Figure 8 (Appendix B). Observe that types near the minimum persist until SE-BME terminates and are estimated accurately, while types with smaller means are eliminated early without precise estimation, reducing evaluations of $w^\star(t)$.

Figure 2(b)-(c) examines how much SE-BME and Simple BME reduce evaluations of $w^\star(t)$ when estimating $\min_{t_n \in \mathcal{T}_n} \mathbb{E}[w^\star(t) \mid t_n]$ for all $n \in \mathcal{N}$ with the accuracy: $\varepsilon = 0.05$ and $\delta = 0.05$. Recall that $K = |\mathcal{T}_n|$ for all $n \in \mathcal{N}$. Exact computation for one player requires $K^N$ evaluations of $w^\star(t)$, which also suffices for all players if values are cached. Since evaluating $w^\star(t)$ via (2) dominates the cost, we focus on unique evaluations (total sample size is reported in Figure 9, Appendix B). SE-BME and Simple BME also benefit from caching. Figure 2 compares unique evaluations for exact computation, $(0.05, 0.05)$-PAC SE-BME, and Simple BME.

Figure 2(b) shows that both SE-BME (solid curves) and Simple BME (dotted curves) require orders of magnitude fewer evaluations than exact computation (dashed curves). While exact computation grows exponentially with $N$ ($K^N$), both SE-BME and Simple BME grow only polynomially—slightly slower than

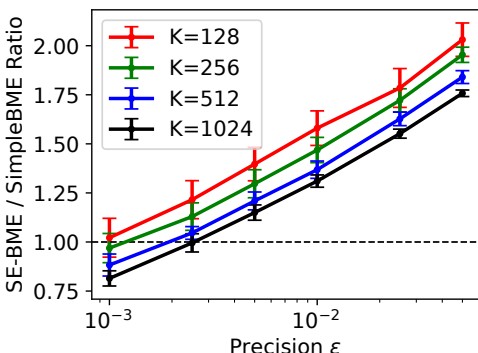

Figure 3: Sample complexity ratio (SE-BME/Simple BME) with $N = 8$ players and $K \in \{128, 256, 512, 1024\}$ types per player. Matching values are transformed to create Bernoulli reward distributions with relatively uniform gap structure. The ratio is averaged over 10 seeds per configuration, and standard deviations are shown with errorbars.

linearly—because the number of players affects reward distribution but not the number of arms. Simple BME achieves slightly better performance than SE-BME across all numbers of players.

Figure 2(c) plots the number of evaluations against $K = |\mathcal{T}_n|$ for any $n \in \mathcal{N}$, with $K$ ranging from 4 to 128. Here, the advantage of SE-BME and Simple BME is smaller since increasing $K$ directly increases the number of arms. Nevertheless, both methods consistently achieve substantial reductions in unique evaluations compared to exact computation.

In the settings of Figure 2, Simple BME has performed slightly better than SE-BME, which is in contrast to what has been observed for Bernoulli reward distributions in Figure 1. We now slightly modify the experimental setting and make it more similar to the one in Figure 1. Specifically, a realized matching value $v$ is transformed to a binary reward $r = \mathbb{1}[v > \tau(K)]$, where $\tau(K) := aK + b$ is the threshold function with $a = -0.757$ and $b = 8.233$ calibrated to make mean rewards relatively uniform[4].

Figure 3 evaluates the ratio of the sample complexity between SE-BME and Simple BME after this transformation. Here, we vary the precision parameter $\varepsilon \in \{0.001, 0.0025, 0.005, 0.01, 0.025, 0.05\}$ and the number of types $K \in \{128, 256, 512, 1024\}$, with confidence level $\delta = 0.05$ and the number of players $N = 8$ fixed. For each configuration, we run SE-BME on 10 random seeds and report the ratio of samples used by SE-BME to the theoretical sample complexity of Simple BME. At $\varepsilon = 0.001$, SE-BME outperforms Simple BME on average for $K \geq 256$, and the advantage of SE-BME becomes statistically significant with $K \geq 512$.

## 8.3 Individual Rationality and Budget Balance with Best Mean Estimation

We next examine how well $\theta$-IR and $\rho$-SBB hold when $\min_{t_n \in \mathcal{T}_n} \{\mathbb{E}[w^\star(t) \mid t_n] - \theta(t_n)\}$ is estimated using BME rather than computed exactly. We use the mechanism design setting from Section 8.2. Recall that $\theta$-IR is guaranteed when $\eta_n$ follows (37), and $\rho$-SBB when it follows (30), but only if $\mathbb{E}[w^\star(t)]$ and $\mathbb{E}[w^\star(t) \mid t_n]$ are computed exactly. Here, we evaluate how well these properties hold when expectations are estimated from samples. Throughout, we set $\rho = 0$ and $\theta \equiv 0$ (i.e., $\theta(t_n) = 0$ for all $t_n \in \mathcal{T}_n$, $n \in \mathcal{N}$).

In Figure 4, we first evaluate $\min_{t_n \in \mathcal{T}_n} \mathbb{E}[w^\star(t) \mid t_n]$ either exactly or via BME, then compute $\eta_n$ using (37) for Columns (a)–(b) and (30) for Columns (c)–(d). We then evaluate the expected utility of each player (left-hand side (LHS) of (8)) in Columns (a) and (c) and the mediator's expected revenue (LHS of (9)) in Columns (b) and (d), setting $h_n(t_{-n}) = \eta_n$ for all $t_{-n} \in \mathcal{T}_{-n}$ and $n \in \mathcal{N}$. Even when the best mean is estimated with BME, these expectations are computed exactly, as they represent actual outcomes. Each

---

[4]The parameters $a = -0.757$ and $b = 8.233$ were obtained by fitting a linear function $\tau(K) = aK + b$ to empirically optimized thresholds across $K \in \{4, 8, 16, 32, 64\}$, where each threshold was selected to minimize the mean squared error between transformed arm means and the ideal evenly-spaced sequence $[1, (K-1)/K, ..., 1/K, 0]$.

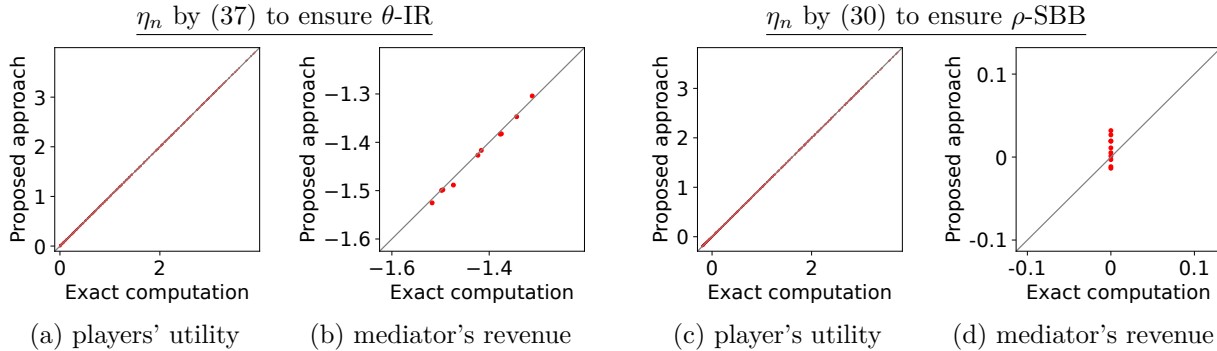

(a) players' utility    (b) mediator's revenue    (c) player's utility    (d) mediator's revenue

Figure 4: The red dots show the expected utility of the players (Columns a, c) or the expected revenue of the mediator (Columns b, d), comparing exact analytical solutions (horizontal axes) with estimates from $(0.05, 0.05)$-PAC SE-BME (vertical axes) for environments with $|\mathcal{N}| = 8$ players, each having $|\mathcal{T}_n| = 8$ possible types. The analytical solution guarantees 0-IR (i.e., $\theta \equiv 0$) in (a)–(b) and 0-SBB (i.e., $\rho = 0$) in (c)–(d). Results are plotted for 10 random seeds; diagonal lines indicate equality.

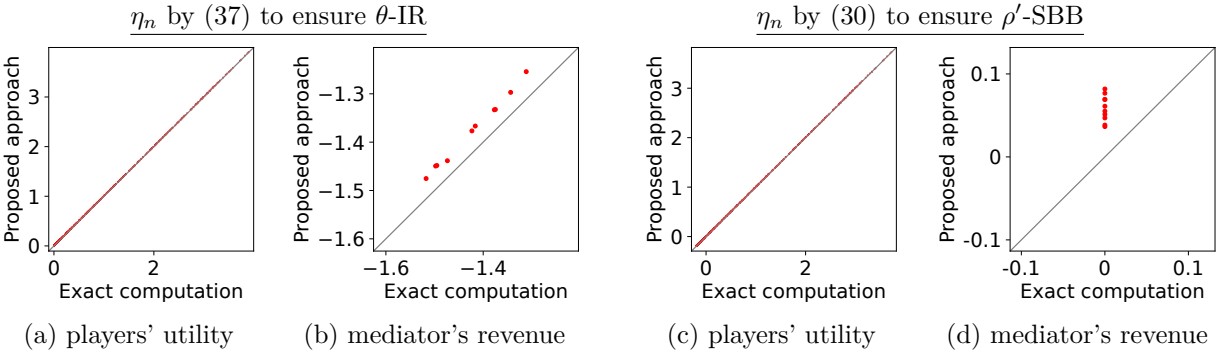

(a) players' utility    (b) mediator's revenue    (c) players' utility    (d) mediator's revenue

Figure 5: Same as Figure 4, except $\rho = 0$ is replaced with $\rho' = 0.05$.

experiment uses 10 random seeds, yielding 10 points in (b) and (d) and $10 \times 8 \times 8 = 640$ points in (a) and (c), one per player-type-seed combination.

Columns (a) and (c) show that players' expected utilities are largely insensitive to whether the best mean is computed exactly (horizontal) or via BME (vertical). Taking a closer look, we can observe that, in this setting, $\theta$-IR with $\theta \equiv 0$ is violated for some players in (c) even with exact computation, while it always holds in (a).

By contrast, Columns (b) and (d) reveal notable differences in mediator revenue, depending on whether the best mean is evaluated with exact computation or with BME. This is expected since (9) aggregates $\sum_{n \in \mathcal{N}} \eta_n$, whereas (8) involves only one $\eta_n$, leading to larger variance in the expected revenue. In (d), 0-WBB (and thus 0-SBB) fails when using BME estimates, though it always holds with exact computation. In (b), 0-WBB fails in both cases, as satisfying 0-WBB and 0-IR for all players (with DE and DSIC) is impossible in this setting.

A simple remedy is to replace $\rho = 0$ with $\rho' > 0$ when computing $\eta_n$ from BME-based estimates of $\min_{t_n \in \mathcal{T}_n} \mathbb{E}[w^\star(t) \mid t_n]$. Using the $(\varepsilon, \delta)$-PAC error bound, $\rho'$ can be chosen to ensure 0-WBB holds with high probability. Similarly, $\theta(t_n) = 0$ can be replaced by $\theta'(t_n) > 0$ to guarantee 0-IR under BME estimates (see Theorem 3).

As an example, we set $\rho' = 0.05$ in Figure 5. The effect of replacing $\rho = 0$ with $\rho' = 0.05$ is as expected: in Columns (b) and (d), mediator revenue under BME (Proposed) shifts upward by $\rho' - \rho = 0.05$. Although less visible in Columns (a) and (c), each player's expected utility shifts left by $(\rho' - \rho)/|\mathcal{N}| = 0.00625$. In practice, $\rho'$ and $\theta'$ can be chosen considering these shifts and the LP feasibility condition (Lemma 1).

## 9    Conclusion

We derived analytical solutions for the LP that ensures DE, DSIC, $\rho$-SBB, and $\theta$-IR. For $N$ players with $K$ types each, the LP has $N K^{N-1}$ variables, while our solution uses only $N$ essential variables. Unlike Osogami et al. (2023), which solves the LP numerically only for $N = K = 2$, we evaluated our solution exactly for $N = K = 8$ (Figure 2). However, it involves a term requiring $K^N$ evaluations of efficient social decisions. We modeled this as best mean estimation in a multi-armed bandit, proposed a PAC estimator, and proved its asymptotic optimality via a sample complexity lower bound. The experiments show that our estimator enables mechanisms for $N = 128$ and $K = 8$ with guaranteed properties.

While the approach marks significant progress, it has limitations that suggest future research directions. First, when types are dependent, the sufficient condition in (10) may not be necessary (Lemma 1), so the LP may remain feasible even if the condition fails. Further work is needed to characterize necessity and to develop efficient methods for dependent types. Second, our mechanisms guarantee SBB and IR only in expectation over type distributions, not *ex post*. While expectation often suffices for risk-neutral agents, mediators and participants should be aware of possible negative realized utilities. Extending the approach to achieve these properties *ex post* is an interesting direction. Finally, our experiments considered up to 128 players and 1024 types—much larger than Osogami et al. (2023) but still limited. Handling environments 10–100 times larger may require improved implementations and resources, while scaling to $10^3$ times larger or continuous type spaces likely needs new ideas. Exploiting structural properties of specific environments could enable scalable mechanism design.

## Acknowledgements

We sincerely thank Junya Honda for identifying an error in the proof presented in a previous version.

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

# A    Details of Section 7

In this section, we prove Lemmas 4-5, which respectively provide bounds on the sample complexity in Theorem 4.

## A.1    Upper Bound

*Proof of Lemma 4.* Since the sample complexity of PAC-BAI in Step 1 is $M$ and Step 2 pulls an arm $m^\star$ times, the sample complexity of Algorithm 2 is $M + m^\star$. Hence, it remains to prove (51).

Recall that $\hat{I}$ is a random variable representing the index of the best arm returned by an $(\varepsilon, \delta)$-PAC BAI. Then we have the following bound:

$$\Pr\left(|\hat{\mu}_{\hat{I}} - \mu_\star| > \frac{3}{2}\varepsilon\right)$$

$$= \Pr\left(\hat{\mu}_{\hat{I}} > \mu_\star + \frac{3}{2}\varepsilon\right) + \Pr\left(\hat{\mu}_{\hat{I}} < \mu_\star - \frac{3}{2}\varepsilon\right) \tag{52}$$

$$\leq \Pr\left(\hat{\mu}_{\hat{I}} > \mu_{\hat{I}} + \frac{3}{2}\varepsilon\right)$$

$$\quad + \Pr\left(\left\{\hat{\mu}_{\hat{I}} < \mu_\star - \frac{3}{2}\varepsilon\right\} \cap \{\mu_{\hat{I}} < \mu_\star - \varepsilon\}\right) + \Pr\left(\left\{\hat{\mu}_{\hat{I}} < \mu_\star - \frac{3}{2}\varepsilon\right\} \cap \{\mu_{\hat{I}} \geq \mu_\star - \varepsilon\}\right) \tag{53}$$

$$\leq \Pr\left(\hat{\mu}_{\hat{I}} > \mu_{\hat{I}} + \frac{3}{2}\varepsilon\right) + \Pr\left(\mu_{\hat{I}} < \mu_\star - \varepsilon\right) + \Pr\left(\left\{\hat{\mu}_{\hat{I}} < \mu_\star - \frac{3}{2}\varepsilon\right\} \cap \{\mu_{\hat{I}} \geq \mu_\star - \varepsilon\}\right) \tag{54}$$

$$\leq \Pr\left(\hat{\mu}_{\hat{I}} > \mu_{\hat{I}} + \frac{3}{2}\varepsilon\right) + \Pr\left(\mu_{\hat{I}} < \mu_\star - \varepsilon\right) + \Pr\left(\hat{\mu}_{\hat{I}} < \mu_{\hat{I}} - \frac{1}{2}\varepsilon\right) \tag{55}$$

$$\leq \Pr\left(\hat{\mu}_{\hat{I}} > \mu_{\hat{I}} + \frac{3}{2}\varepsilon\right) + \delta + \Pr\left(\hat{\mu}_{\hat{I}} < \mu_{\hat{I}} - \frac{1}{2}\varepsilon\right), \tag{56}$$

where the last inequality follows from PAC$(\varepsilon, \delta)$ of BAI.

Since $\hat{\mu}_{\hat{I}}$ is the average of $m^\star$ samples from arm $\hat{I}$, applying Hoeffding's inequality to the last term of (56), we obtain

$$\Pr\left(\hat{\mu}_{\hat{I}} < \mu_{\hat{I}} - \frac{1}{2}\varepsilon\right) = \sum_{k \in [1,K]} \Pr\left(\hat{\mu}_k < \mu_k - \frac{1}{2}\varepsilon \,\middle|\, \hat{I} = k\right) \Pr(\hat{I} = k) \tag{57}$$

$$\leq \sum_{k \in [1,K]} \exp\left(-2\left(\frac{1}{2}\varepsilon\right)^2 m^\star\right) \Pr(\hat{I} = k) \tag{58}$$

$$= \exp\left(-2\left(\frac{1}{2}\varepsilon\right)^2 m^\star\right), \tag{59}$$

where the inequality is obtained by applying Hoeffding's inequality to the sample mean $\hat{\mu}_k$ of $m^\star$ independent random variables having support in $[0, 1]$. We can also show the following inequality in an analogous manner:

$$\Pr\left(\hat{\mu}_{\hat{I}} > \mu_{\hat{I}} + \frac{3}{2}\varepsilon\right) \leq \exp\left(-2\left(\frac{3}{2}\varepsilon\right)^2 m^\star\right). \tag{60}$$

By applying (59)-(60) to (56), we finally establish the bound to be shown:

$$\Pr\left(|\hat{\mu}_{\hat{I}} - \mu_\star| > \frac{3}{2}\varepsilon\right) \le \delta + \exp\left(-2\left(\frac{3}{2}\varepsilon\right)^2 m^\star\right) + \exp\left(-2\left(\frac{1}{2}\varepsilon\right)^2 m^\star\right) \tag{61}$$

$$\le \delta + \exp\left(-2\left(\frac{3}{2}\varepsilon\right)^2 \frac{2}{\varepsilon^2}\log\frac{1.22}{\delta}\right) + \exp\left(-2\left(\frac{1}{2}\varepsilon\right)^2 \frac{2}{\varepsilon^2}\log\frac{1.22}{\delta}\right)$$

$$\text{by the definition of } m^\star \tag{62}$$

$$= \delta + \left(\frac{\delta}{1.22}\right)^9 + \frac{\delta}{1.22} \tag{63}$$

$$\le \left(1 + \frac{1}{1.22^9} + \frac{1}{1.22}\right)\delta \tag{64}$$

$$\le 2\,\delta. \tag{65}$$

$\square$

Notice that the naive approach of sampling each arm $\Theta((1/\varepsilon^2)\log(1/\delta))$ times, which also trivially falls within the upper bound in Theorem 4, would only guarantee that the best mean is estimated with the error bound of $\varepsilon$ with probability at least $(1-\delta)^K$. Conversely, it would require sampling each arm $\Omega((1/\varepsilon^2)\log(K/\delta))$ times to obtain the same error bound with probability $1 - \delta$.

### A.2 Lower Bound

We derive the lower bound in Lemma 5 by reducing BME to the $\varepsilon$-Biased Coin Problem:

**Definition 2** ($\varepsilon$-Biased Coin Problem). *For $0 < \varepsilon < 1$, consider a Bernoulli random variable $B$ whose mean $\alpha$ is known to be either $\alpha^+ \coloneqq (1+\varepsilon)/2$ or $\alpha^- \coloneqq (1-\varepsilon)/2$. The $\varepsilon$-Biased Coin Problem asks to correctly identify whether $\alpha = \alpha^+$ or $\alpha = \alpha^-$.*

A lower bound on the sample complexity for solving the $\varepsilon$-Biased Coin Problem is known as the following lemma, for which we provide a proof for completeness:

**Lemma 6** (Chernoff (1972); Even-Dar et al. (2002)). *For $0 < \delta < 1/2$, any algorithm that solves the $\varepsilon$-Biased Coin Problem correctly with probability at least $1 - \delta$ has expected sample complexity at least $\Omega((1/\varepsilon^2)\log(1/\delta))$.*

*Proof.* Although the lemma is stated in Even-Dar et al. (2002) with reference to Chernoff (1972), this specific lemma is neither stated nor proved explicitly in Chernoff (1972). For completeness, here, we prove the lemma following the general methodology provided in Chernoff (1972). Specifically, we derive the expected sample size required by the sequential probability-ratio test (SPRT; Section 10 of Chernoff (1972)), whose optimality (Theorem 12.1 of Chernoff (1972)) will then establish the lemma.

Consider two hypotheses, $\theta_1$ and $\theta_2$, for the probability distribution $\mathbb{P}$ of a random variable $X$, which takes either the value of 1 or $-1$, where

$$\mathbb{P}(X = 1 \mid \theta_1) = \frac{1+\varepsilon}{2} \tag{66}$$

$$\mathbb{P}(X = 1 \mid \theta_2) = \frac{1-\varepsilon}{2} \tag{67}$$

Consider the SPRT procedure that takes i.i.d. samples, $X_1, X_2, \ldots, X_N$, from $\mathbb{P}$ until the stopping time $N$ when

$$\lambda_N \coloneqq \prod_{n=1}^{N} \frac{\mathbb{P}(X_n \mid \theta_1)}{\mathbb{P}(X_n \mid \theta_2)} = \prod_{n=1}^{N}\left(\frac{\varepsilon+1}{\varepsilon-1}\right)^{X_n} \tag{68}$$

hits either $A \in \mathbb{R}$ or $1/A$. When $\lambda_N$ hits $A$, we identify $\theta_1$ as the correct hypothesis. When $\lambda_N$ hits $1/A$, we identify $\theta_2$ as the correct hypothesis.

Let

$$S_N := \log \lambda_N = \sum_{n=1}^{N} X_n \log \frac{1+\varepsilon}{1-\varepsilon}. \tag{69}$$

Since $N$ is a stopping time, by Wald's lemma, we have

$$\mathbb{E}[S_N \mid \theta_1] = \mathbb{E}[N \mid \theta_1] \, \mathbb{E}[X \mid \theta_1] \log \frac{1+\varepsilon}{1-\varepsilon} \tag{70}$$

$$= \mathbb{E}[N \mid \theta_1] \, \varepsilon \log \frac{1+\varepsilon}{1-\varepsilon} \tag{71}$$

$$\mathbb{E}[S_N \mid \theta_2] = -\mathbb{E}[N \mid \theta_2] \, \varepsilon \log \frac{1+\varepsilon}{1-\varepsilon}. \tag{72}$$

Let $\delta$ be the probability of making the error in identifying the correct hypothesis. Then we must have

$$\mathbb{E}[S_N \mid \theta_1] = (1-\delta) \log A + \delta \log(1/A) \tag{73}$$

$$\mathbb{E}[S_N \mid \theta_2] = \delta \log A + (1-\delta) \log(1/A). \tag{74}$$

By (70)-(74), we have

$$\mathbb{E}[N \mid \theta_1] = \mathbb{E}[N \mid \theta_2] = \frac{1-2\,\delta}{\varepsilon \log \frac{1+\varepsilon}{1-\varepsilon}} \log A. \tag{75}$$

Now, notice that $S_N$ hits $\log A$ when we have

$$|\{n : X_n = 1\}| - |\{n : X_n = -1\}| \geq \frac{\log A}{\log \frac{1+\varepsilon}{1-\varepsilon}} \tag{76}$$

for the first time and hits $-\log A$ when we have

$$|\{n : X_n = -1\}| - |\{n : X_n = 1\}| \geq \frac{\log A}{\log \frac{1+\varepsilon}{1-\varepsilon}} \tag{77}$$

for the first time. Hence, by the gambler's ruin probability, we have

$$\delta = \frac{1 - \left(\frac{1+\varepsilon}{1-\varepsilon}\right)^{\frac{\log A}{\log \frac{1+\varepsilon}{1-\varepsilon}}}}{1 - \left(\frac{1+\varepsilon}{1-\varepsilon}\right)^{2\frac{\log A}{\log \frac{1+\varepsilon}{1-\varepsilon}}}} = \frac{1}{1 + \left(\frac{1+\varepsilon}{1-\varepsilon}\right)^{\frac{\log A}{\log \frac{1+\varepsilon}{1-\varepsilon}}}}, \tag{78}$$

which implies

$$A = \frac{1-\delta}{\delta}. \tag{79}$$

Plugging the last expression into (75), we obtain

$$\mathbb{E}[N \mid \theta_1] = \mathbb{E}[N \mid \theta_2] = \frac{1-2\,\delta}{\varepsilon \log \frac{1+\varepsilon}{1-\varepsilon}} \log \frac{1-\delta}{\delta} = O\left(\frac{1}{\varepsilon^2} \log \frac{1}{\delta}\right). \tag{80}$$

$\square$

*Proof of Lemma 5.* We will construct an algorithm that correctly identifies the mean $\alpha$ of $B$ with probability at least $\delta$ using at most $M/K$ samples of $B$ in expectation given the access to an $(\varepsilon/2, \delta/2)$-PAC BME with sample complexity $M$. First, the algorithm independently draws $i^+$ and $i^-$ from the uniform distribution over $[1, K]$.

Consider two environments of $K$-armed bandit, $\mathcal{E}^+$ and $\mathcal{E}^-$, where every arm gives the reward according to the Bernoulli distribution with mean $\alpha^- = (1 - \varepsilon)/2$ except that the reward of arm $i^+$ in $\mathcal{E}^+$ has the same distribution as $B$ and the reward of arm $i^-$ in $\mathcal{E}^-$ has the same distribution as $1 - B$. Note that the algorithm can simulate the reward with the known mean $\alpha^-$ using the algorithm's internal random number generators. Only when the algorithm pulls arm $i^+$ of $\mathcal{E}^+$ or arm $i^-$ of $\mathcal{E}^-$, it uses the sample of $B$, which contributes to the sample complexity of the algorithm.

The algorithm then runs two copies of the $(\varepsilon/2, \delta/2)$-PAC BME with sample complexity $M$ in parallel: one referred to as BME$^+$ is run on $\mathcal{T}^+$, and the other referred to as BME$^-$ is run on $\mathcal{T}^-$. At each step, let $M^+$ be the number of samples BME$^+$ has taken from arm $i^+$ by that step and $M^-$ be the corresponding number BME$^-$ has taken from arm $i^-$. If $M^+ < M^-$, the algorithm lets BME$^+$ pull an arm; otherwise, the algorithm lets BME$^-$ pull an arm. Therefore, $|M^+ - M^-| \le 1$ at any step.

This process is continued until one of BME$^+$ and BME$^-$ terminates and returns an estimate $\hat{\mu}$ of the best mean. If BME$^+$ terminates first, then the algorithm determines that $\alpha = \alpha^-$ if $\hat{\mu} < 1/2$ and that $\alpha = \alpha^+$ otherwise. If BME$^-$ terminates first, then the algorithm determines that $\alpha = \alpha^+$ if $\hat{\mu} < 1/2$ and that $\alpha = \alpha^-$ otherwise. Due to the $(\varepsilon/2, \delta/2)$-PAC property of BME$^+$ and BME$^-$, the algorithm correctly identifies the mean of $B$ with probability at least $1 - \delta$. Formally, if $\alpha = \alpha^+$, we have

$$
\begin{aligned}
\Pr\left(\hat{\mu} < \frac{1}{2}\right) &= \Pr\left(\hat{\mu} < \frac{1}{2} \,\middle|\, \text{BME}^+ \text{ terminates first}\right) \Pr(\text{BME}^+ \text{ terminates first}) \\
&\quad + \Pr\left(\hat{\mu} < \frac{1}{2} \,\middle|\, \text{BME}^- \text{ terminates first}\right) \Pr(\text{BME}^- \text{ terminates first}) \qquad (81) \\
&\le \frac{\delta}{2} + \frac{\delta}{2} \qquad\qquad\qquad\qquad\qquad\qquad\qquad\qquad\qquad\qquad\qquad\qquad (82) \\
&= \delta. \qquad\qquad\qquad\qquad\qquad\qquad\qquad\qquad\qquad\qquad\qquad\qquad\qquad (83)
\end{aligned}
$$

Analogously, $\Pr\left(\hat{\mu} < \frac{1}{2}\right) \le \delta$ can be shown if $\alpha = \alpha^-$.

What remains to prove is the sample complexity of the algorithm. Recall that each of BME$^+$ and BME$^-$ pulls arms at most $M$ times before it terminates due to their sample complexity. Notice that the arms in $\mathcal{E}^-$ are indistinguishable when $\alpha = \alpha^-$, and the arms in $\mathcal{E}^+$ are indistinguishable when $\alpha = \alpha^+$. Therefore, at least one of BME$^+$ and BME$^-$ is run on the environment where the arms are indistinguishable. Since $i^-$ and $i^-$ are sampled uniformly at random from $[1, K]$, BME (either BME$^+$ or BME$^-$) would take at most $M/n$ samples from $B$ in expectation if the arms are indistinguishable. Since $|M^+ - M^-| \le 1$, we establish that the sample complexity of the algorithm is $O(M/n)$ in expectation. $\qquad\square$

# B  Details of Section 8

In this section, we provide the pseudo-code of Successive Elimination for Best Arm Identification ($(\varepsilon, \delta)$-PAC SE-BAI), proofs of the correctness of SE-BME and SE-BAI, and additional experimental results, including the details of computational resources needed in our experiments in Section 8. For completeness, we also provide standard correctness proofs of SE-BME and SE-BAI.

**Proposition 3.** *Algorithm 3 is an $(\varepsilon, \delta)$-PAC BME.*

---

**Algorithm 4** Successive Elimination for Best Arm Identification (($\varepsilon, \delta$)-PAC SE-BAI)

---

**Require:** $\varepsilon, \delta$
1: Let $\mathcal{R} \leftarrow \{1, \ldots, K\}$ be the set of remaining arms
2: Let $t \leftarrow 0; \alpha \leftarrow 1$
3: **while** $|\mathcal{R}| > 1$ and $\alpha > \frac{\varepsilon}{2}$ **do**
4:   Pull each arm $k \in \mathcal{R}$ once and update the sample average $\hat{\mu}_k$
5:   $t \leftarrow t + 1$
6:   $\alpha \leftarrow \sqrt{\frac{1}{2t} \log \left( \frac{\pi^2 K t^2}{6 \delta} \right)}$
7:   **for all** $k \in \mathcal{R}$ **do**
8:     Remove $k$ from $\mathcal{R}$ if $\max_{\ell \in \mathcal{R}} \hat{\mu}_\ell - \hat{\mu}_k \geq 2\alpha$
9:   **end for**
10: **end while**
11: **return** $\underset{k \in \mathcal{R}}{\operatorname{argmax}} \hat{\mu}_k$

---

*Proof.* Let $\mu_\star := \max_{k \in [1,K]} \mu_k$ be the best mean, $\hat{\mu}$ be the best mean estimated by Algorithm 3, and $\hat{\mu}_k^{(t)}$ be the average of the first $t$ samples from arm $k$. Let $\alpha_t := \sqrt{\frac{1}{2t} \log \left( \frac{\pi^2 K t^2}{3 \delta} \right)}$. Then we have

$$\Pr(|\hat{\mu} - \mu_\star| \leq \varepsilon)$$

$$\geq \Pr(\text{At every iteration, the error in the estimated mean is less than } \alpha_t \text{ for any arm in } \mathcal{R}.^5) \tag{84}$$

$$= \Pr \left( \bigcap_{t=1}^{\infty} \bigcap_{k \in [1,K]} \{|\{\hat{\mu}_k^{(t)} - \mu_k| < \alpha_t\} \right) \tag{85}$$

$$= 1 - \Pr \left( \bigcup_{t=1}^{\infty} \bigcup_{k \in [1,K]} \{|\{\hat{\mu}_k^{(t)} - \mu_k| \geq \alpha_t\} \right) \tag{86}$$

$$\geq 1 - \sum_{t=1}^{\infty} \sum_{k \in [1,K]} \Pr \left( \{|\{\hat{\mu}_k^{(t)} - \mu_k| \geq \alpha_t\} \right) \quad \text{by union bound} \tag{87}$$

$$\geq 1 - 2K \sum_{t=1}^{\infty} \exp \left( -2\alpha_t^2 t \right) \quad \text{by Hoeffding's inequality} \tag{88}$$

$$= 1 - \delta \frac{6}{\pi^2} \sum_{t=1}^{\infty} \frac{1}{t^2} \quad \text{by definition of } \alpha_t \tag{89}$$

$$= 1 - \delta. \tag{90}$$

$\square$

**Proposition 4.** *Algorithm 4 is an ($\varepsilon, \delta$)-PAC BAI.*

---

[5]This condition suffices because it ensures that the best arms always remain in $\mathcal{R}$.

*Proof of Proposition 4.* Let $\alpha_t := \sqrt{\log\left(\frac{\pi^2 \, K \, t^2}{6 \, \delta}\right)/(2 \, t)}$. Let $\mathcal{B}$ be the set of the (strictly) best arms. Let $\hat{\mu}_k^{(t)}$ be the average of the first $t$ samples from arm $k$. Then we have

$$\text{Pr}(\text{Algorithm 4 selects an } \varepsilon\text{-best arm.})$$

$$\geq \text{Pr}(\text{At every iteration, all arms in } \mathcal{B} \text{ remain in } \mathcal{R} \text{ and any arm in } \mathcal{R} \text{ is } 2\alpha_t\text{-best.}^6) \tag{91}$$

$$\geq \text{Pr}\left(\bigcap_{t=1}^{\infty} \bigcap_{k\in\mathcal{B}} \{\hat{\mu}_k^{(t)} > \mu_k - \alpha_t\} \bigcap_{\ell\notin\mathcal{B}} \{\hat{\mu}_\ell^{(t)} < \mu_\ell + \alpha_t\}\right) \tag{92}$$

$$= 1 - \text{Pr}\left(\bigcup_{i=1}^{\infty} \bigcup_{k\in\mathcal{B}} \{\hat{\mu}_k^{(t)} \leq \mu_k - \alpha_t\} \bigcup_{\ell\notin\mathcal{B}} \{\hat{\mu}_\ell^{(t)} \geq \mu_\ell + \alpha_t\}\right) \tag{93}$$

$$\geq 1 - \sum_{t=1}^{\infty} \left(\sum_{k\in\mathcal{B}} \text{Pr}(\hat{\mu}_k^{(t)} \leq \mu_k - \alpha_t) + \sum_{\ell\notin\mathcal{B}} \text{Pr}(\hat{\mu}_\ell^{(t)} \geq \mu_\ell + \alpha_t)\right) \quad \text{by union bound} \tag{94}$$

$$\geq 1 - \sum_{t=1}^{\infty} K \exp(-2 \, \alpha_t^2 \, t) \quad \text{by Hoeffding's inequality} \tag{95}$$

$$= 1 - \delta \frac{6}{\pi^2} \sum_{t=1}^{\infty} \frac{1}{t^2} \quad \text{by the definition of } \alpha_t \tag{96}$$

$$= 1 - \delta. \tag{97}$$

$\square$

## B.1 Best Arm Identification and Best Mean Estimation

Here, we compare Successive Elimination for BAI and BME. Specifically, we evaluate $(\varepsilon, \delta)$-PAC Successive Elimination for BME (SE-BME; Algorithm 3) against its BAI counterpart (SE-BAI; Algorithm 4).

SE-BME and SE-BAI differ in three ways. First, SE-BME stops when $\alpha < \varepsilon$ (vs. $\varepsilon/2$ for SE-BAI) because it can return an overestimate of a suboptimal arm if within $\varepsilon$ of the best mean. Second, SE-BAI terminates when one arm remains, as it need not estimate its mean precisely. Finally, SE-BAI uses smaller $\alpha_t$ since it only needs to account for one-sided errors (underestimation for the best arm, overestimation for others).

Figure 6 reports total sample sizes for SE-BME and SE-BAI across varying $\varepsilon$ and arm counts $K$. Arms have Bernoulli rewards with equally spaced means ($\mu_k = (k - 0.5)/K$ for $k = 1, \ldots, K$). Each point averages 10 runs; standard deviations are too small to be visible. Overall, SE-BME generally requires fewer samples than SE-BAI, except when $K$ is small (large mean gaps). This efficiency of SE-BAI in large-gap settings is intuitive, as identifying the best arm requires less precise estimation.

## B.2 Comprehensive Comparison of Best Mean Estimation Algorithms

As discussed in Section 8.1, we compare four BME algorithms across varying numbers of arms $K$, accuracy parameters $\varepsilon$, and confidence levels $\delta$. All algorithms solve the same BME problem and are compared under identical parameter settings:

1. **SE-BME** (Algorithm 3): Successive Elimination tailored directly for BME, which eliminates arms with low estimated means while continuing to estimate the remaining arms until convergence.

2. **SE**: Successive Elimination for BAI (Algorithm 4) combined with Algorithm 2 to adapt it for BME. This approach first identifies the best arm, then estimates its mean.

---

[6]This condition suffices because the algorithm stops either when $|\mathcal{R}| = 1$ or when $\alpha_t \leq \varepsilon/2$, which implies that only $\varepsilon$-best arms are in $\mathcal{R}$ when the algorithm stops.

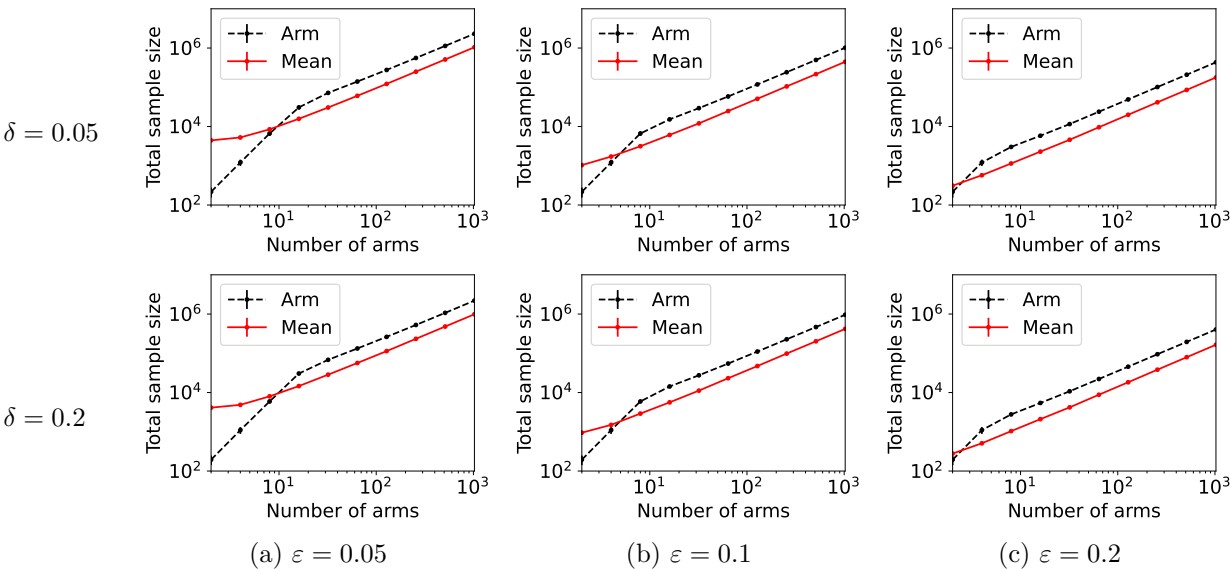

Figure 6: Total sample sizes required by SE-BME (Mean; Algorithm 3) and SE-BAI (Arm; Algorithm 4) for Bernoulli rewards with equally spaced means, across varying $\varepsilon$ and $\delta$ values.

3. **UGapEc**: The UGapEc algorithm (Gabillon et al., 2012) for BAI combined with Algorithm 2. UGapEc is an $(\varepsilon, \delta)$-PAC BAI having an asymptotically optimal sample complexity.

4. **Simple**: A simple baseline that collects $m = \lceil (2\varepsilon^2)^{-1} \log(2K/\delta) \rceil$ samples from each arm independently, computes the sample mean for each arm, and returns the maximum sample mean[7]. The total sample complexity is $O(K/\varepsilon^2 \log(K/\delta))$, only a $\log(K)$ factor worse than the asymptotically optimal complexity.

Figure 7 presents a comprehensive comparison across nine parameter combinations. SE-BME consistently achieves the lowest sample complexity across all settings, demonstrating the benefit of direct BME design over BAI-based adaptations. The Simple baseline performs competitively, particularly for small $K$ (left columns) and large $\varepsilon$ (bottom rows), validating its $O(K/\varepsilon^2 \log(K/\delta))$ complexity as reasonable in practice. Both SE and UGapEc require more samples than SE-BME, with their performance gap widening as $K$ increases (moving from left to right in each row). The trends are consistent across different values of $\delta$ (columns within each row), indicating that the relative performance of these algorithms is robust to changes in the confidence parameter.

Figure 9 shows the total sample size, rather than the unique sample size, required by exact computation (dashed curves) and SE-BME (solid curves). The total sample size for exact computation is $N K^N$. While $w^\star(t)$ is evaluated $N$ times for each $t$ in exact computation, SE-BME may waste evaluating the same $w^\star(t)$ more often particularly when there are only a small number of players. This reduces benefits of SE-BME for total sample size, as compared to the unique sample size.

In Figure 10, we show the Root Mean Squared Error (RMSE) in the expected utility of each player (a) and the expected revenue of the mediator (b) that are estimated with BME for the case with $|\mathcal{N}| = 8$ players, each with $|\mathcal{T}_n| = 8$ types. Here, we fix $\delta = 0.05$ and vary $\varepsilon$ from 1.0 to 0.15 in the BME. For each pair of $(\delta, \varepsilon)$, the experiments are repeated 10 times with different random seeds. The total sample size increases as the value of $\varepsilon$ decreases. Hence, the purple dots correspond to $\varepsilon = 1.0$, and the red dots are $\varepsilon = 0.15$. Overall, we can observe that RMSE can be reduced by using small $\varepsilon$ at the expense of increased sample size,

---

[7]By Hoeffding's inequality and a union bound, $\lceil (2\varepsilon^2)^{-1} \log(2K/\delta) \rceil$ samples per arm ensure uniform $\varepsilon$-accurate mean estimation with probability $1 - \delta$. This yields an $\varepsilon$-accurate estimate of the best mean, even though it does not necessarily identify the true best arm, which may be $2\varepsilon$-suboptimal.

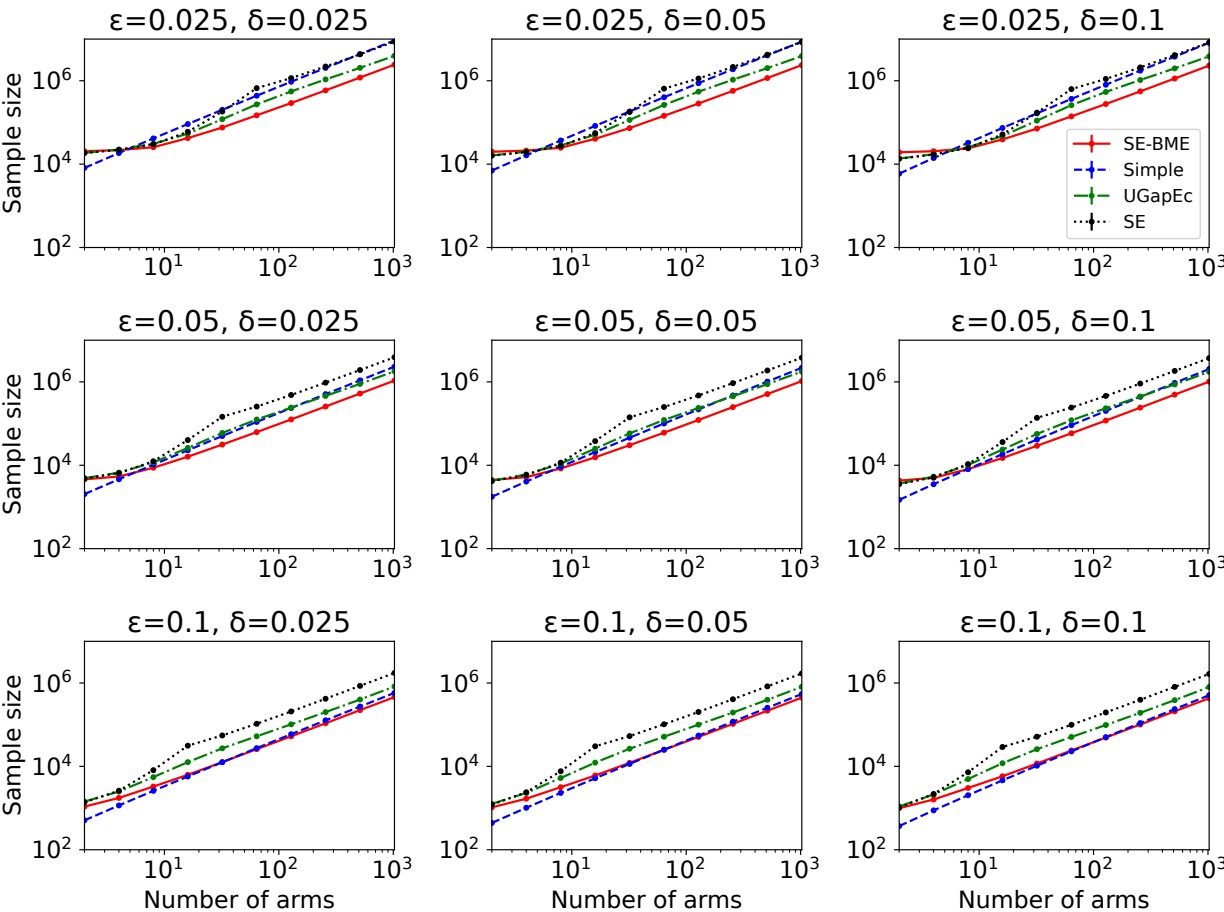

Figure 7: Total sample sizes required by four BME algorithms (SE-BME, Simple, UGapEc, SE) for Bernoulli rewards with equally spaced means ($\mu_k = (k-0.5)/K$ for $k = 1, \ldots, K$) across varying $\varepsilon$ and $\delta$ values. Each point averages 10 runs; standard deviations are too small to be visible.

that relatively large values such as $\varepsilon = 0.5$ gives reasonably small RMSE, and that larger values of $\varepsilon$ have diminishing effects on RMSE.

We have run all the experiments on a single core with at most 64 GB memory without GPUs in a cloud environment. The associated source code is submitted as a supplementary material and will be open-sourced upon acceptance. Table 1 summarizes the CPU time and maximum memory require to generate each figure. For example, CPU time for Figure 6(a) is the time to generate three panels in Column (a) of Figure 6. Note that the CPU time and maximum memory reported in Table 1 are not optimized and include time and memory for storing intermediate results and other processing for debugging purposes; these should be understood as the computational requirements to execute the source code as is.

## C  Societal Impacts

We expect that the proposed approach has several positive impacts on trading networks in particular and the society in general. In particular, the proposed approach enables mechanisms that can maximize the efficiency of a trading network and minimize the fees that the participants need to pay to the mediator. Also, the DSIC guaranteed by the proposed approach would make it more difficult for malicious participants to manipulate the outcome of a trading network.

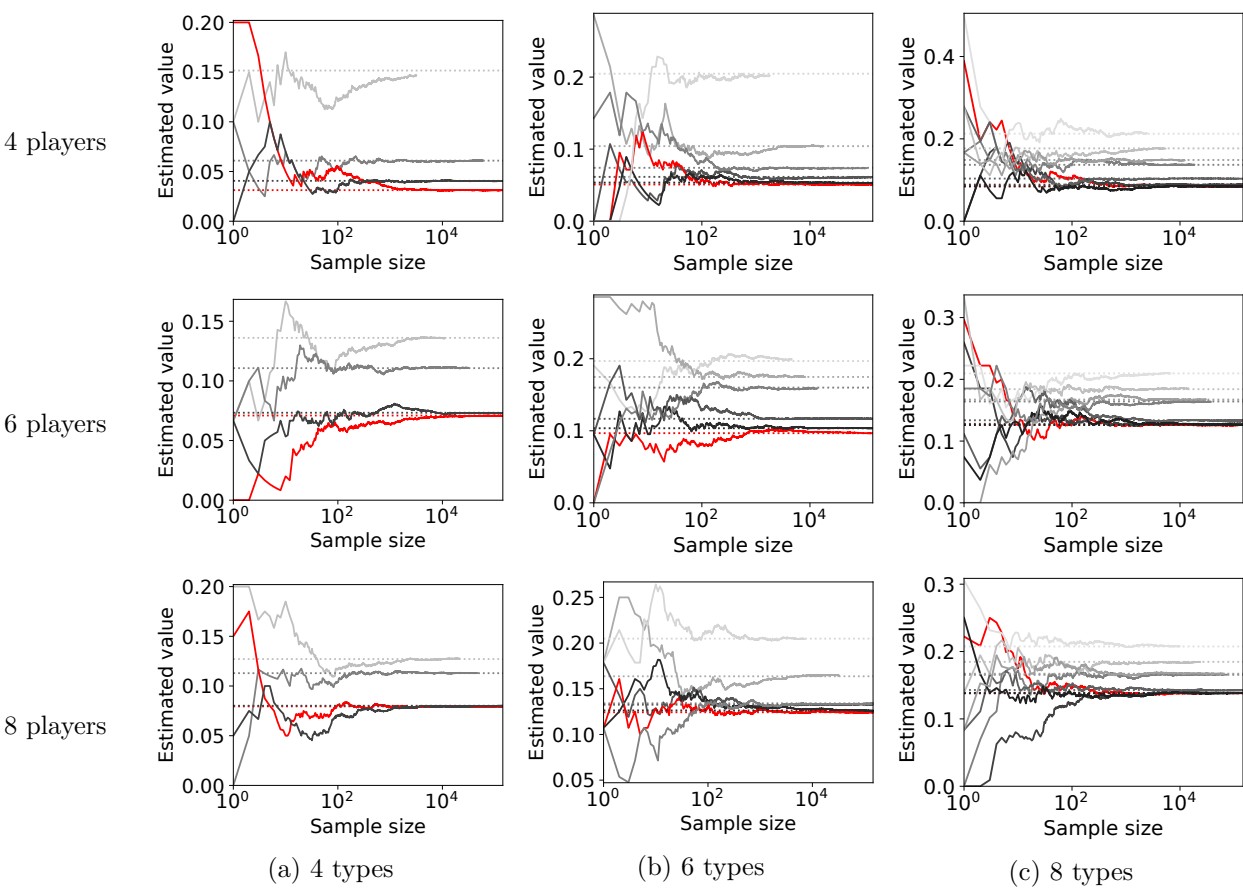

(a) 4 types          (b) 6 types          (c) 8 types

Figure 8: Representative sample paths showing estimated values of $\mathbb{E}[w^\star(t) \mid t_n]$ for $t_n \in \mathcal{T}_n$ versus sample size under $(0.01, 0.1)$-PAC SE-BME with varying number of players and types.

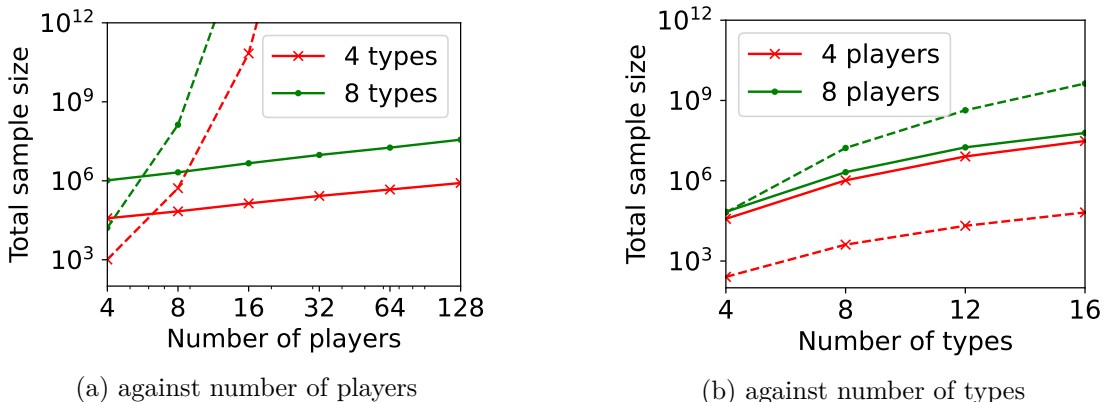

(a) against number of players          (b) against number of types

Figure 9: The total sample size (the number of $t$ which $w^\star(t)$ is evaluated with) required by the exact computation of $\min_{t_n \in \mathcal{T}_n} \mathbb{E}[w^\star(t) \mid t_n], \forall n \in \mathcal{N}$ (dashed curves) and by $(0.25, 0.1)$-PAC SE-BME (solid curves).

On the other hand, the proposed approach might have negative impacts depending on where and how it is applied. For example, although the proposed approach guarantees individual rationality, some participants might benefit less from the mechanism designed with our approach than other participants. This can happen, because maximizing the social welfare does not mean that all the participants are treated fairly.

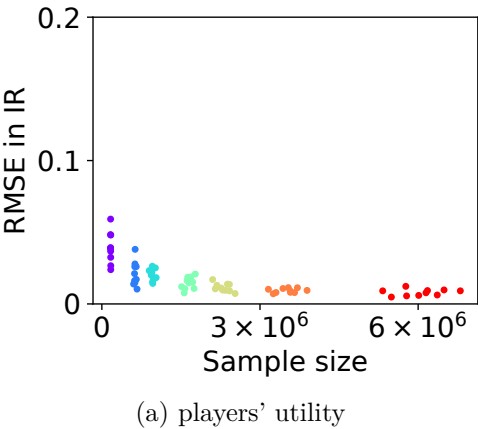

(a) players' utility

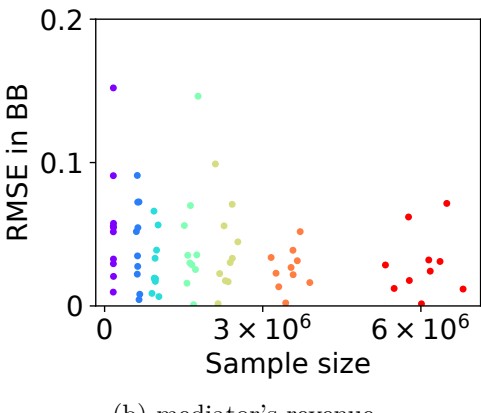

(b) mediator's revenue

Figure 10: Root mean squared error in (a) the expected utility of each player and (b) the expected revenue of the mediator against the total sample size, when there are $|\mathcal{N}| = 8$ players, each with $|\mathcal{T}_n| = 8$ possible types. Here, we set $\delta = 0.05$ and vary $\varepsilon$ from 1.0 (purple), 0.5, 0.4, 0.3, 0.25, 0.2, to 0.15 (red).

Table 1: CPU time and maximum memory required to generate figures

| Figure | CPU Time (seconds) | Max Memory (GB) |
|---|---|---|
| Figure 6(a) | 413.1 | < 1 |
| Figure 6(b) | 81.9 | < 1 |
| Figure 6(c) | 40.4 | < 1 |
| Figure 8(a)[†] | 0.7 | < 1 |
| Figure 8(b)[†], Figure 2(a) | 1.1 | < 1 |
| Figure 8(c)[†] | 80.1 | 1.9 |
| Figure 2(b) and Figure 9(a) | 7,883.0 | 65.5 |
| Figure 2(c) and Figure 9(b) | 2,935.2 | 17.0 |
| Figure 4(a)-(b) and Figure 5(a)-(b)[‡] | 17,531.6 | 1.6 |
| Figure 4(c)-(d) and Figure 5(c)-(d)[‡] | 17,416.3 | 1.6 |
| Figure 10 | 527.9 | < 1 |

[†] Figure 8 shows the results with one random seed, but here the CPU Time reports the average over 10 seeds, and Max Memory reports the maximum over 10 seeds.

[‡] Figure 5 could have been obtained by simply reusing and shifting Figure 4, but here the CPU time reports the time to generate the two figures without reuse.

Before applying the mechanisms designed with the proposed approach, it is thus recommended assessing whether such fairness needs to be considered and to take any actions that mitigate the bias if needed.

