# OpenReview forum: "Achieving PAC Guarantees in Mechanism Design through Multi-Armed Bandits"
_TMLR — Accepted by TMLR_

### Review · Reviewer_U9X2 · 2025-12-27

**Summary Of Contributions:**

This paper studies a general mechanism design problem with a mediator: $n$ players submit their private types to a mediator, who then determines a social outcome and every player's payment to the mediator. The payment can be negative, meaning that the mediator pays the player.  Desired properties for the mechanism are:

* DE: decision efficiency -- the outcome should maximize social welfare;
* DSIC: dominant strategy incentive compatible;
* (ex-ante) $\theta$-IR: the expected utility of every player is larger than $\theta \ge 0$ -- a generalization of ex-ante 0-IR;
* (ex-ante) $\rho$-Weak (or Strong) Budget Balance: The total expected payment to the mediator is at least (equal to) $\rho \ge 0$.

Importantly, Myerson-Satterthwaite Impossibility Theorem implies that DE + DSIC + interim IR + ex-ante WBB mechanisms do not exist.  This paper relaxes interim IR to ex-ante IR, thus obtaining some positive results.  In particular, the paper's main contributions are:

* The authors formulate the optimal mechanism design problem as a linear program, and provide a sufficient condition under which the LP is feasible (hence a desired mechanism exists).  The authors also prove that the condition is necessary when players' types are independent.
* Under the sufficient condition, the optimal mechanism can actually be implemented by a VCG mechanism with a constant pivot rule: each player's payment $\tau_n(t)$ is equal to a constant function (independent of the player's own type and others' types) minus other players' total welfare under the socially-optimal decision.
* The sufficient condition, as well as the VCG pivot rule, is closely related to the quantity $\kappa_n(\theta) := \min_{t_n \in T_n}  E[ w^{\star}(t) | t_n] - \theta(t_n) $, which requires evaluating the expected optimal social welfare $E[w^{\star}(t) | t_n]$ conditioning on player $n$'s type $t_n$.  Evaluating this quantity exactly requires exponential time.  To address this issue, the authors:
  * (Section 6) draw a connection to the Best-Mean-Estimation (BME) problem in Mult-Armed Bandit (MAB), treating each type $t_n$ as an arm and the expected optimal social welfare $E[w^*(t) | t_n]$ as mean reward;
  * (Section 7) then, characterize the tight sample complexity for the BME problem;
  * (Section 8) and apply some BME algorithms to evaluating $\kappa_n(\theta)$, experimentally demonstrating the effectiveness over the exponential-time exact evaluation method.



### Key strength

* The authors demonstrate that relaxing interim IR to ex-ante IR circumvents Myerson-Satterthwaite Impossibility Theorem, leading to the existence of efficient, DSIC, ex-ante IR, and ex-ante BB mechanisms (under some conditions).  The found condition is both sufficient and necessary in the important special case of independent types.  These observations might be interesting to the Mechanism Design people in the TMLR audience.

### Key Weakness

* Some comparison benchmarks in the experimentss are not appropriate.  They do not support the claims well enough.  See the following part.

**Additional Comments:**

(I fixed some typos in my initial reviews)

**Audience:**

Yes

**Audience Explanation:**

Yes. People in Game Theory, in particular those who use Machine Learning methods to solve Mechanism Design problems, might be interested.

**Broader Impact Concerns:**

Due to its mostly theoretical nature, this work doesn't seem to have major ethical concerns.  If any, the impact might be positive because the paper aims to find mechanisms that maximize social welfare.

**Claims And Evidence:**

No

**Claims Explanation:**

I will first mention some contributions that are well supported.

(1) A key contribution of the work is to demonstrate that, once we relax interim IR to ex-ante IR, Myerson-Satterthwaite Impossibility Theorem can be circumvented.  In particular, under some conditions, optimal mechanisms (DE, DSIC, ex-ante IR, ex-ante BB) exist and can even be implemented by VCG mechanism with a constant pivot rule.  This is an interesting result that exploits ex-ante IR.  And this result is supported by theoretical proof, which I believe is correct.

(2) The tight sample complexity of the BME problem (arising from the mechanism design problem) is also supported by rigorous proofs.



I will then mention some claims that are not well supported.

(3) Section 8.1 aims to study the empirical performance of BME algorithms with moderate number of arms (instead of the asymptotic sample complexity with large number of arms), and wants to claim the advantage of the SE-BME algorithm (Successive Elimination for BME, Algorithm 3) with moderate number of arms.  This purpose is not met, I think, because the authors compare the SE-BME algorithm and a BAI algorithm (Successive Elimination for Best Arm Identification, Algorithm 4), under the same accuracy parameter $\epsilon$.  These two algorithms are for different problems (BME and BAI), so this comparison is not meaningful.  Achieving $\epsilon$-accuracy in BME has a different meaning from achieving $\epsilon$-accuracy in BAI.  It is not meaningful to compare the sample complexities of algorithms for different problems under the same $\epsilon$ parameter.

A more meaningful comparison should be: Plug the Successive Elimination BAI algorithm into Algorithm 2 to obtain an "SE-BAI-instantiated" BME algorithm, and compare this BME algorithm with the SE-BME algorithm (Algorithm 3), and compare them under the same required accuracy $\epsilon$ to see whose sample complexity is smaller.  In this way, we are comparing different algorithms for the same problem (BME problem), which meets the purpose of this subsection.

(4) In Section 8.2, the authors claim that their BME method (SE-BME, Algorithm 3) reduces the computational cost significantly.  To support this claim, they compare SE-BME with the exact computation method (Section 8.2).  However, I think a more appropriate comparison benchmark should be the straightforward BME algorithm that, for each arm, collects $O(1/\epsilon^2 \log(K/\delta))$ samples to estimate its mean, and then outputs the best estimated mean (the sample complexity comes from a Hoeffding's inequality combined with a union bound over $K$ arms).  The sample complexity of this straightforward algorithm should be $O(K/\epsilon^2 \log(K/\delta))$, which is only worse than the tight sample complexity $O(K/\epsilon^2 \log(1/\delta))$ by a factor of $\log(K)$.  I would appreciate the authors to compare SE-BME with this straightforward BME algorithm as well, in order to really demonstrate the advantage of SE-BME.

**Requested Changes:**

Besides the (3) and (4) I mentioned in the "Are the claims supported by..." part, I suggest the following additional changes:

(Suggestion 1) In Equation (11), change the $t$ and $t_n$ inside min to be $t'$ and $t_n'$ to indicate that they are not part of the input $t_{-n}$, but are new samples from $P$, and mention that $h_n(t_{-n})$ is independent of $t_{-n}$.  I got confused by the notations initially and realized their meanings only after seeing Lemma 2.

(Suggestion 2) Proof of Lemma 2.  In the end, the authors claim "The necessity follows in exactly the same way as the proof of the necessary condition in Lemma 1".  Lemma 1 shows that condition (10) is necessary for the main LP (7)-(9) to be feasible when types are independent, while Lemma 2 wants to show that condition (10) is necessary for the existene of an optimal solution with a constant pivot rule (regardless of type independence or not).  The claims of Lemmas 1 and 2 are different, so I don't immediately see how the proof of Lemma 1 also applies to Lemma 2.  I do believe that condition (10) is necessary for the claim in Lemma 2, but I think a different proof should be written.   This issue is not critical to my recommendation.

(Suggestion 3) The citation about Myerson-Satterthwaite Impossibility Theorem in the related work section, "no mechanism can guarantee ex-post DE (decision efficiency), DSIC, IR, and WBB", is not accurate.  The Myerson-Satterthwaite result is actually stronger: according to their Theorem 3, no mechanism can guarantee ex-post DE + Bayesian-IC + interim IR + ex-ante WBB.  That implies that no mechanism can guarantee ex-post DE + DSIC + interim IR + ex-ante WBB, which is stronger than the authors' citation about the MS result.  The authors consider ex-post DE + DSIC + ex-ante IR + ex-ante WBB in this paper, so the important distinction with the MS result is ex-ante IR.  This distinction should be highlighted.

---

> ### Author Response · Authors · 2026-02-16
>
> We thank the reviewer for their thorough and constructive feedback. Below we address each point and suggestion in detail.
>
> ================================================================================
> Response to Point (3): Comparison of SE-BME and SE-BAI algorithms
> ================================================================================
>
> We thank the reviewer for correctly pointing out that for evaluating algorithmic performance, we should compare different algorithms that all solve the same BME problem. We have now revised Section 8.1 to include this more meaningful comparison. In addition to SE and UGapEc, which are constructed with Algorithm 2 from respective BAI algorithms, we include Simple BME, which is what the reviewer has suggested in Point (4).  The new comparison in Section 8.1 supports the advantage of SE-BME over these baselines.
>
> Our original comparison between SE-BME and SE-BAI (now presented in Appendix B.1, Figure 6) was intended to illustrate the fundamental difference between the BME and BAI problems themselves—specifically, how achieving ε-accuracy has different meanings in these two settings and how this affects sample complexity. We have kept these results in the appendix as they provide some insight into the problem formulations.
>
> ================================================================================
> Response to Point (4): Comparison with simple BME algorithm
> ================================================================================
>
> We thank the reviewer for suggesting this important baseline comparison. We have noticed that the original results were generated with a loose estimate on the support of the reward distributions, which resulted in unnecessarily large sample size for the proposed approach.  In the revision, we have also tightened the support estimate; hence, the number of evaluations of the proposed approach has changed.  This fix has also enabled smaller and more realistic values of $\varepsilon$ and $\delta$. We have now added the comparison with these correction to Figure 2 in Section 8.2.
>
> The results show that both SE-BME and Simple BME have substantial advantages over exact computation, while Simple BME slightly outperforms SE-BME in this particular setting.
>
> To further investigate relative advantages of SE-BME and Simple BME in mechanism design settings, we have conducted additional experiments whose results are summarized in Figure 3.  Here, the rewards are transformed to Bernoulli distributions, and we vary the number of types $K$ and the precision parameter $\varepsilon$.  The results show that SE-BME outperforms Simple BME when $K$ is sufficiently large and $\varepsilon$ is sufficiently small.
>
> ================================================================================
> Response to Suggestion 1: Notation clarity in Equation (11)
> ================================================================================
>
> We thank the reviewer for pointing out this potential source of confusion. We have revised the notation and added a clarifying sentence immediately after Equation (11).
>
> ================================================================================
> Response to Suggestion 2: Proof of Lemma 2 necessity
> ================================================================================
>
> We thank the reviewer for this careful observation. We have updated the statement of Lemma 2 to make it more precise. With this updated statement, the necessity can be shown with a simple argument, as shown in the revision.
>
> ================================================================================
> Response to Suggestion 3: Myerson-Satterthwaite citation
> ================================================================================
>
> We thank the reviewer for this careful reading of the Myerson-Satterthwaite result. The reviewer is correct that the original MS Theorem 3 proves a stronger impossibility result with Bayesian-Nash IC (which is weaker than DSIC), making their impossibility more general. We added a footnote explicitly noting this point.
>
> Please notice that we assume interim IR similar to Meyerson-Satterthwaite, which is made clearer in Introduction (please see the changes in Page 2 as well as the definition of $\theta$-IR in Page 4).  We relax IR and WBB by introducing parameters $\theta$ and $\rho$ to exactly characterize when the mechanisms with desired properties exist.

---

### Review · Reviewer_mrrd · 2026-01-11

**Summary Of Contributions:**

Motivated by settings like trading networks, the paper studies mechanism design for general multi-agent environments where a mediator (platform) chooses a centralized outcome to improve social welfare. The authors focuses an ``open platform'' goal: allocate efficiently while avoiding deficits or surpluses, in contrast to profit-maximizing intermediaries that may extract most surplus.

Formally, they move beyond single-sided auctions to general environments (e.g., double-sided auctions, matching markets, trading networks) and aim to design mechanisms with four target properties: decision efficiency (DE), dominant-strategy incentive compatibility (DSIC), strong budget balance (SBB) with a revenue target $\rho$, and individual rationality (IR) with type-dependent threshold $\theta(\cdot)$. A key modeling choice is that SBB and IR are required only in expectation over types, while DE and DSIC must hold ex post for every realized type profile.

On results, they (i) derive an analytical class of optimal solutions to a linear program for automated mechanism design, showing that it can be represented via “essential variables” whose number is exponentially smaller than the original LP’s variables, but (ii) identify a remaining computational bottleneck: exactly evaluating a key term still needs exponentially many evaluations as $N$ grows. They address this by (iii) casting that evaluation as a multi-armed bandit best-mean estimation problem and giving an $(\epsilon, \delta)$-PAC estimator with asymptotically optimal sample complexity, which reduces the computational burden enough to scale experiments up to $N=128$ players.

**Additional Comments:**

N/A

**Audience:**

Yes

**Audience Explanation:**

The paper may be of interest to algorithmic game theory and learning community.

**Claims And Evidence:**

Yes

**Claims Explanation:**

The authors have provided rigorous Lemma/Theorem statements, and also provide complete and rigorous proofs.

**Requested Changes:**

The title emphasizes “multi-armed bandits,” which initially led me to expect an online learning component with an exploration–exploitation tradeoff. In the current manuscript, however, the bandit machinery is used primarily as a pure-exploration / best-mean estimation subroutine to reduce the computational burden of estimating certain expectations, rather than to study sequential decision-making with exploitation. I suggest re-positioning this contribution more explicitly, either by clarifying early (title/abstract/introduction) that the role of bandits is as a statistical estimation tool, or by adjusting the title framing to avoid implying an online learning/exploitation aspect, before the publication of this work.

---

> ### Author Response · Authors · 2026-02-16
>
> We thank the reviewer for their constructive feedback regarding the framing of the multi-armed bandit machinery in our work.
>
> ================================================================================
> Response to Requested Change: Clarifying the role of multi-armed bandits
> ================================================================================
>
> We have revised the manuscript to clarify the role of bandits early and prominently.  In Abstract and Introduction, we now explicitly state that our use of MAB is for pure exploration to estimate expectations rather than online learning with exploration-exploitation tradeoffs.
>
> We believe these changes address the reviewer's concern about potential misunderstanding of our contribution. We have kept the title as is because "multi-armed bandits" accurately describes the technical framework we employ, and the early clarifications now prevent misconceptions about online learning or exploitation. However, we are open to feedback on whether the current clarifications in the abstract and introduction are sufficient, or whether a title adjustment would be preferable.

---

### Review · Reviewer_TkDS · 2026-02-11

**Summary Of Contributions:**

This paper studies a specific class of mechanisms for a general market where agents may either buy or sell items. The authors introduce a very general market model in which agents may be both sellers and buyers, and list natural desiredata on mechanisms, namely: DSIC, IR, budget balance (BB), and decision efficiency (DE), and then decide to focus on the family of VCG mechanisms.

The authors build on a previous LP formulation of the problem and provide a sufficient and necessary condition for the LP to admit solutions (the condition is necessary only for independent distributions). Then, they find an analytical characterization of solutions.

This analytic solution involves computing objects that are the min of the expected value of a random variable (which depends on the efficiency of the optimal solution under a given type). The authors then propose a MAB approach to estimate such minimum.

Strenghts:
- the model studied is fairly general
- the problem is within the topics of interest for TMLR

Weakneses
- the LP formulation is mainly from a previous paper, and the analytical manipulations are fairly standard
- the idea of estimating the min using online learning is actually nice, but I feel like the authors are overcomplicating it quite a lot.
- the authors do not explain why they focus on the family of VCG mechanisms. Is it without loss of generality?

**Additional Comments:**

The contribution of the paper is fairly restricted, building on previous techniques and with a suboptimal writing.

I like the idea of estimating the pivot parameters via online learning, but I think that the authors should definitely improve the writing and provide a more complete comparison with other approaches (e.g., what about studying the dual LP? what about other learning techniques? Why bandits and not expert algorithms?)

**Audience:**

Yes

**Audience Explanation:**

Provided that the authors improve the presentation of the paper as detailed below, I think that the idea of estimating the min with online learning is a cute one that may be interesting in general

**Claims And Evidence:**

Yes

**Claims Explanation:**

The theoretical results are backed by formal proofs that look sound.

I did not check the math in detail.

**Requested Changes:**

- Consider adding an Our Result section, presenting clearly what is the problem that you are solving and your results. Also using theorem statement
- there is no need to introducing all the machinery in Section 6 to estimate (41), just present your algorithm and state its properties
- Please address the comments below

---

> ### Author Response · Authors · 2026-02-16
>
> We thank the reviewer for their constructive feedback. Below we address weaknesses and requested changes.
>
> ================================================================================
> Response to Weakness: Why focus on VCG mechanisms?
> ================================================================================
>
> We have added a footnote in Section 4 clarifying this choice. Since our setting involves finite type spaces, VCG mechanisms may not exhaust all mechanisms satisfying the four properties (DE, DSIC, $\theta$-IR, $\rho$-SBB) for a given $(\theta,\rho)$. Nevertheless, we restrict attention to VCG mechanisms, as they are both of theoretical interest in the literature and of practically importance with various real-world implementations. Focusing on this principled class provides a tractable path to analytical solutions while maintaining strong theoretical guarantees.
> (made minor edit on Feb. 17 for clarity)
>
> ================================================================================
> Response to Requested Change 1: Add "Our Results" section
> ================================================================================
>
> We have added a dedicated "Our Results" subsection in the Introduction (Section 1) that clearly presents:
> - The problem we solve
> - Two main theorems (informal versions with Lemma/Theorem references)
> - Key contributions highlighted in separate paragraphs with complexity analysis
>
> This makes our contributions immediately visible to readers scanning the paper.
>
> ================================================================================
> Response to Requested Change 2: Simplify Section 6
> ================================================================================
>
> We have made Section 6 more accessible by:
> - Moving detailed BAI vs BME distinguishing examples to a footnote
> - Adding a new paragraph (immediately after the footnote) explaining the section structure and key results and providing a "skip to" option directing readers to Proposition 2 for the final sample complexity result
>
> We believe the technical content is necessary to establish that our approach provides a PAC guarantee, but the improved organization makes it easier for readers to navigate based on their interests.
> (made minor edit on Feb. 17 for clarity)
>
> ================================================================================
> Response to Requested Change 3: Comparisons with other approaches
> ================================================================================
>
> We address the reviewer's specific questions about alternative approaches:
>
> **Dual LP**: The dual of our LP formulation also has exponential size. Our analytical characterization (Lemma 2) reduces the problem to $N$ essential variables (constant pivot rules), bypassing the need to solve either primal or dual explicitly. The remaining computational challenge is evaluating the expectations $\min_{t_n}\mathbb{E}[w^\star(t)\mid t_n]$ in the analytical solution, which the dual does not address.
>
> **Other learning techniques**: We now provide more comprehensive experimental comparisons in Section 8 by adding two baselines, **Simple BME** and **BME with UGapEc**.
> (made minor edit on Feb. 17 for clarity)
>
> **Expert algorithms**: Classical experts algorithms (e.g., weighted majority, Hedge) require full feedback (observing rewards for all arms at each time step) which would require evaluating $w^\star(t)$ for all $K$ type profiles per round, defeating the purpose of our approach. While bandit-feedback variants like EXP3 exist that only require evaluating the chosen arm, these algorithms are still designed for a fundamentally different objective: minimizing cumulative regret over time in a sequential decision-making setting. In contrast, our goal is pure exploration to estimate a specific quantity ($\min_{t_n}\mathbb{E}[w^\star(t)\mid t_n]$) with minimal sample complexity and PAC guarantees. BME is the natural MAB formulation for this estimation task.

---

### Author Response · Authors · 2026-02-17
**Minor issues fixed**

Dear reviewers,

Thank you very much for your thoughtful and constructive feedback.  We have now provided our responses to all of your requested changes and other imprtant comments.  We have also uploaded a new pdf.

We noticed that the our responses and the pdf provided on Feb. 16th had some minor issues, which we have corrected those uploaded a new version on Feb. 17th.  The updates are minor, but we would appreciate it if you could refer to the latest version for clarity.

---

### Decision · Action_Editor_gMBn · 2026-04-04

**Recommendation:** Accept as is

**Additional Comments:**

Although I have selected "Accept as is", Reviewer U9X2 has some easy-to-implement writing suggestions for the authors. These suggestions are not visible to the authors, so I have copied them below. Please implement these suggestions.

1. Section title capitalization should be unified: for example, the title of Section 8 "Numerical experiments" should be capitalized; the title of Section 8.2 "Effectiveness of Best Mean Estimation in mechanism design" should be fixed.

2. Page 3: "Under these conditions, the Myerson-Satterthwaite Impossibility Theorem (Myerson & Satterthwaite, 1983) shows that no mechanism can guarantee ex post DE, DSIC, IR, and WBB in all such environments, unlike VCG auctions." I would say "... no mechanism can guarantee ex post DE, DSIC, IR, and WBB in general environments, unlike the VCG auctions in the single-sided environment".

3. Section 8.2, Page 15: In Figure 2(a), the best arm is the arm with the highest reward. But the BME problem for mechanism design aims to estimate the minimal value $\\min\_{t\_n} \\mathbb{E} [ w^\ast(t) \\mid t\_n ]$ . I saw that the authors wrote "To apply bandit algorithms, we normalize $w^\ast(t)$ and switch the minimization problem to maximization", but the next paragraph still says "minimum" instead of "maximum". I'd suggest flipping Figure 2(a), so the best arm is the minimal arm, and say something like "We use bandit algorithms to solve the minimization problem $\\min\_{t\_n} \\mathbb{E} [ w^\ast(t) \\mid t\_n ]$ (instead of the maximization problem as in Section 8.1)", and then continue to use "minimum" in the remaining paragraphs.

4. Page 16: There is a typo in the sentence "Specifically, a realized matching value v is transformed to a binary reward r = ⊮[v > τ (K)]".

**Audience:**

Yes

**Audience Explanation:**

Yes. Specifically, one reviewer commented that the authors' idea of estimating the minimum using online learning may be interesting to the community. Another reviewer commented positively:
> As Myerson-Satterthwaite has shown that DE, DSIC, interim IR, ex-ante WBB mechanisms do not exist in general mechanism design problems, the LP in this paper can not always have a feasible solution. One of the contributions of this paper is to give a sufficient (and necessary for independent types) condition under which this paper's LP has a feasible solution with a VCG format.

This paper is sufficiently interesting, especially to the mechanism design community, to warrant publication.

**Claims And Evidence:**

Yes

**Claims Explanation:**

I am unsure how closely proofs were checked, but all reviewers seem to be satisfied with the statement of theoretical guarantees and the rigor of the formal proofs. Upon request by one reviewer, the authors improved the experimental comparisons. The reviewer who asked for better experimental comparison is now satisfied.